# Bayesian Logistic Regression Model for Sub-Areas

**Lu Chen** [1,*] and **Balgobin Nandram** [2]

1   National Institute of Statistical Sciences, 1750 K Street NW Suite 1100, Washington, DC 20006, USA
2   Worcester Polytechnic Institute, Stratton Hall 103, Worcester, MA 01609, USA
*   Correspondence: lchen@niss.org

**Abstract:** Many population-based surveys have binary responses from a large number of individuals in each household within small areas. One example is the Nepal Living Standards Survey (NLSS II), in which health status binary data (good versus poor) for each individual from sampled households (sub-areas) are available in the sampled wards (small areas). To make an inference for the finite population proportion of individuals in each household, we use the sub-area logistic regression model with reliable auxiliary information. The contribution of this model is twofold. First, we extend an area-level model to a sub-area level model. Second, because there are numerous sub-areas, standard Markov chain Monte Carlo (MCMC) methods to find the joint posterior density are very time-consuming. Therefore, we provide a sampling-based method, the integrated nested normal approximation (INNA), which permits fast computation. Our main goal is to describe this hierarchical Bayesian logistic regression model and to show that the computation is much faster than the exact MCMC method and also reasonably accurate. The performance of our method is studied by using NLSS II data. Our model can borrow strength from both areas and sub-areas to obtain more efficient and precise estimates. The hierarchical structure of our model captures the variation in the binary data reasonably well.

**Keywords:** hierarchical Bayesian model; integrated nested normal approximation; MCMC; metropolis sampler; numerical integration; parallel computing

## 1. Introduction

The Nepal Living Standard Survey (NLSS) II (see [1]) is a two-stage stratified sampling. A random sample of wards (areas) were selected from six strata and 12 households (sub-areas) were selected from each sampled ward. All individuals in each sampled household were interviewed. One interest is on health status, a binary variable. To make smooth estimates of the finite population proportion of the individuals with good health in each household, we focus on hierarchical Bayesian (HB) models with sub-area random effects to obtain reliable "indirect" estimates for numerous small areas or sub-areas. Most of the sample surveys are designed to provide reliable "direct" estimates of interests for large areas or domains (e.g., state level, national level). However, direct estimates are not reliable for areas or domains for which only small samples or no samples are available—see [2].

In many applications, some areas, e.g., states and wards, are sampled; in each sampled area, a sample of sub-areas, e.g., counties and households, is further selected. Ref. [3] proposed a one-fold hierarchical Bayesian logistic regression model and applied the model to NLSS II data. The main objective is to make an inference for the finite population proportion of individuals with a specific character for each area. However, the one-fold model ignores the sub-area level structure in the data. As an extension of [3], we are particularly interested in small area models that can capture the hierarchical structure of the NLSS II data in this paper. Although the one-fold basic models are very popular and in common use in producing reliable estimates, the hierarchical structure of the data and the consistency between the estimates for different levels may not hold. In particular, the sampling designs of many population-based surveys were two-stage stratified sampling as

NLSS II. But if we use a one-fold unit level model to fit the data, the sub-area level effects will have been ignored. Ref. [4] studied the case that the data follow a normal model with a two-stage (three-stage) hierarchical structure, while the fitted model has a one-stage (two-stage) hierarchical structure using posterior predictive *p*-values. Ref. [5] discussed the ability to detect a three-stage model when a two-stage model is actually fitted.

Two-fold models are an important extension of basic small area models. Many authors have considered the problems and proposed these kinds of models. Much of the literature focuses on continuous data. Ref. [6] proposed a sub-area level model which provides model-based estimates that account for the hierarchical structure of data. Two-fold sub-area level models were studied by [2,7–9], and many others. This type of model is an area-level model which extends the Fay-Harriot model (see [10]) to the sub-area level. Two-fold nested error regression models were considered by [11,12]. On the other hand, some literature focus on the categorical data. Ref. [13] described a HB model to make an inference about the finite population proportion under two-stage clustering sampling. Ref. [14] extended the Beta-Binomial model to the two-fold model and used Gibbs sampling to obtain the posterior estimates. Ref. [15] showed that the two-fold Beta-Binomial model is preferable over the one-fold one if the data have a hierarchical structure. Ref. [16] extended [15] to accommodate heterogeneous correlations. They used a HB model to make a posterior inference about the finite population proportion of each area, accounting for intracluster correlations. Ref. [17] discussed the sub-area Beta-Binomial model and applied the model to estimate the finite population proportion of healthy individuals in each household covered by the NLSS II, assuming no covariate was available.

Bayesian logistic regression models with random effects are suitable for handling binary data with covariates. Ref. [18] discussed discrimination between the logit and the complementary log-log link functions by using the logistic regression model. Roberts, Ref. [19], discussed logistic regression for the sample survey data (not small area estimation). Ref. [20] showed how to accelerate the Gibbs sampler for a model with latent variables introduced earlier by [21] for Bayesian probit analysis. Ref. [22], discussed the logistic regression model by using the empirical Bayesian approach. Ref. [23] showed how to analyze binary data with covariates to maintain conjugacy for both the logistic and Poisson regression models. The analysis of binary data with covariates under nonignorable nonresponse was discussed by [24]. Ref. [3] proposed a hierarchical Bayesian logistic regression model for binary data in a small area estimation. This model is a unit level model without a sub-area effect. Our two-fold sub-area model is an extension of this logistic regression model. We add the sub-area level random effect into the model which can capture the hierarchical structure of the sampled data. At the same time, we add more hyper-parameters into the model, which make the inference more complicated. However, we propose an approximation method called the integrated nested normal approximation (INNA), which solves the difficulties.

The other side of our application is that there are numerous small areas (households and individuals) and MCMC methods, which involve complicated integrals, and cannot handle them efficiently. "Big data" are defined as data that are too big to comfortably process on a single machine [25]. The researchers considered consensus Monte Carlo methods that split the data across several machines. They proposed algorithms that perform distributed approximate Bayesian analyses in order to minimize the communication between computers. The parallel MCMC methods for non-Gaussian posterior distributions were discussed by [26]. Fortunately, in survey sampling, the design generally uses a stratification which is not artificial, and, in this case, consensus Monte Carlo may not be needed; it will be a good idea for a large stratum.

The integration involved in Bayesian inference is usually intractable, which is true for our logistic regression model. The approximation techniques are desired. The procedure we used to approximate the posterior density of the parameters of the logistic regression sub-area model, INNA, is similar to the integrated nested Laplace approximation (INLA) originally proposed by [27], but they are actually different. INLA is a quite popular algo-

rithm and an alternative to MCMC for big data analysis if the joint posterior density is very complicated. It requires posterior modes, and, for numerous small areas, the computation of modes becomes time-consuming and challenging for the logistic regression model or any generalized linear mixed models. Yet, INLA has found many useful applications, such as in Poisson regression by [28], and in spatial point pattern data by [29]. We note that INLA can be problematic, especially for logistic and Poisson hierarchical regression models, even if the modes can be computed. Ref. [30], attempted to improve INLA using a copula-based correction, which adds complexity to INLA. Our approximation method, INNA, which does not require finding posterior modes, uses a sampling-based procedure accommodated by the multiplication rule of probability. Instead of finding the posterior modes, INNA finds the approximate modes in closed form, facilitated by the empirical logistic transform ([31]) and the second-order Taylor series approximation.

On the other hand, two-fold models can capture the heterogeneity between samples within not only areas but also sub-areas. Many model-based estimation techniques for the sampling variances have been considered in the literature, but most of them for the area-level model: see [32–34].

In Section 2, a full description of a sub-area HB logistic regression model is given. In Section 3, we describe the integrated nested normal approximation (INNA) computation method and some theoretical results are provided. The exact MCMC method is presented in Appendix A. The exact method refers to MCMC methods without further approximation. In Section 4, we apply the model to the NLSS II data to provide smoothed estimates of the household proportions of members in good health for both sampled and nonsampled households. Some comparisons between INNA and the exact method are presented. Finally, in Section 5, we make concluding remarks and discuss the future work.

## 2. Sub-Area Logistic Regression Model

In this section, we discuss the sub-area HB logistic regression model at the unit level. In the NLSS II data, we have binary data (good health versus poor health) for each individual within a household, and these households are within wards. The observations are available at the unit level and so is the reliable auxiliary information. However, the model and method we proposed for small areas and sub-areas is not only for this application on NLSS II data. It can be also applied to other population-based surveys with binary responses which contain small areas or/and sub-areas.

Suppose that there are $L$ small areas (wards) in the finite population and that, within the $i^{\text{th}}$ area, there are $N_i$ sub-areas (households). Within the $j^{\text{th}}$ sub-area, there are $M_{ij}$ individuals. We assume that $\ell(< L)$ areas are sampled and a simple random sample of $n_i(< N_i)$ households is taken from the $i^{\text{th}}$ area. All individuals in the sampled households are sampled. Here, we assume the survey weights are the same within all households in each area. Actually, the design is almost self-weighting.

Let $y_{ijk}, \; k = 1, \ldots, m_{ij}, j = 1, \ldots, n_i, i = 1, \ldots, \ell$ denote the binary responses. Let $\underset{\sim}{y} = (y_{ijk}, \; k = 1, \ldots, m_{ij}, j = 1, \ldots, n_i, i = 1, \ldots, \ell)'$. Let $y_{ij} = \sum_{k=1}^{m_{ij}} y_{ijk}$ be the number with response 1 and $m_{ij}$ be the total number of people who responded. Let $\underset{\sim}{x}_{ijk} = (1, x_{ijk1}, \ldots, x_{ijkp})'$ be the $(p + 1)$ vector with $p$ covariates for individuals and an intercept.

We use $P$ to represent the population proportion and $p$ as the sample proportion. Let $p_{ij}$ be the corresponding sample probability of $y_{ij}, j = 1, \ldots, n_i, i = 1, \ldots, \ell$.

The primary interests are the finite population proportions of the households, which are $P_{ij} = \frac{1}{M_{ij}} \sum_{k=1}^{M_{ij}} y_{ijk}, \; j = 1, \ldots, N_i, \; i = 1, \ldots, \ell$ and the finite population proportions of the areas, which are $P_i = \frac{1}{N_i} \sum_{j=1}^{N_i} \sum_{k=1}^{M_{ij}} y_{ijk}, \; i = 1, \ldots, \ell$.

In the content of the logistic regression model, the two-fold hierarchical Bayesian logistic regression model for the sub-area means, $\mu_{ij}$, is

$$y_{ijk}|\underset{\sim}{\beta}, v_i, \mu_{ij} \overset{ind}{\sim} \text{Bernoulli}\left\{\frac{e^{x'_{ijk}\underset{\sim}{\beta}+v_i+\mu_{ij}}}{1+e^{x'_{ijk}\underset{\sim}{\beta}+v_i+\mu_{ij}}}\right\}, k = 1, \ldots, m_{ij},$$

$$\mu_{ij}|\sigma^2 \overset{iid}{\sim} \text{Normal}(0, \sigma^2), j = 1, \ldots, n_i, \tag{1}$$

$$v_i|\delta^2 \overset{iid}{\sim} \text{Normal}(0, \delta^2), i = 1, \ldots, \ell,$$

$$\pi(\underset{\sim}{\beta}, \delta^2, \sigma^2) \propto \frac{1}{(1+\delta^2)^2}\frac{1}{(1+\sigma^2)^2}, \delta^2 > 0, \ \sigma^2 > 0.$$

Here, $\mu_{ij}$, $i = 1, \ldots, \ell; j = 1, \ldots, n_i$ are the sub-area level random effects, which are not in the area-level model in [3]. $v_i$, $i = 1, \ldots, \ell$ are the area random effects and $\underset{\sim}{\beta} = (\beta_0, \beta_1, \ldots, \beta_p)'$ are the regression coefficients, with $\sigma^2, \delta^2$ as the variance of the random effects, respectively.

In order to apply our approximation method and make an inference for posterior distribution, we use an equivalent model.

First, we separate $\underset{\sim}{\beta}$ into $\beta_0$ and $\underset{\sim}{\beta}_{(0)}$, where $\underset{\sim}{\beta}_{(0)} = (\beta_1, \beta_2, \ldots, \beta_p)^T$. We set $\beta_0$ as the mean of $\underset{\sim}{v}$, and then we can omit the intercept term from the covariate $x_{ijk}$. Second, we introduce a new parameter, $w_{ij} = v_i + \mu_{ij}$, in order to set $v_i$ and $\mu_{ij}$ independently and make it easy to make an inference for both of them. We have

$$y_{ijk}|w_{ij}, \underset{\sim}{\beta}_{(0)} \overset{ind}{\sim} \text{Bernoulli}\left\{\frac{e^{x'_{ijk}\underset{\sim}{\beta}_{(0)}+w_{ij}}}{1+e^{x'_{ijk}\underset{\sim}{\beta}_{(0)}+w_{ij}}}\right\}, k = 1, \ldots, m_{ij},$$

$$w_{ij}|v_i, \sigma^2 \overset{ind}{\sim} \text{Normal}(v_i, \sigma^2), j = 1, \ldots, n_i, \tag{2}$$

$$v_i|\beta_0, \delta^2 \overset{iid}{\sim} \text{Normal}(\beta_0, \delta^2), i = 1, \ldots, \ell,$$

$$\pi(\underset{\sim}{\beta}, \delta^2, \sigma^2) \propto \frac{1}{(1+\delta^2)^2}\frac{1}{(1+\sigma^2)^2}, \delta^2 > 0, \ \sigma^2 > 0.$$

The joint posterior density for the parameters is

$$\pi(\underset{\sim}{v}, \underset{\sim}{w}, \underset{\sim}{\beta}, \sigma^2, \delta^2|\underset{\sim}{y}) \propto \prod_{i=1}^{\ell}\prod_{j=1}^{n_i}\prod_{k=1}^{m_{ij}}\left[\frac{e^{(x'_{ijk}\underset{\sim}{\beta}_{(0)}+w_{ij})y_{ijk}}}{1+e^{x'_{ijk}\underset{\sim}{\beta}_{(0)}+w_{ij}}}\right] \times \left(\frac{1}{\sqrt{2\pi\sigma^2}}\right)^n \exp\left\{-\sum_{i=1}^{l}\sum_{j=1}^{ni}\frac{(w_{ij}-v_i)^2}{2\sigma^2}\right\}$$

$$\times \left(\frac{1}{\sqrt{2\pi\delta^2}}\right)^l \exp\left\{-\sum_{i=1}^{l}\frac{(v_i-\beta_0)^2}{2\delta^2}\right\}\frac{1}{(1+\sigma^2)^2}\frac{1}{(1+\delta^2)^2}. \tag{3}$$

The posterior density is a non-standard multivariate density, and there are difficulties in fitting it using MCMC methods, more so when $n_i$, $m_{ij}$ are large. This motivates our approximate methods.

## 3. Integrated Nested Normal Approximation Method

In this section, we discuss the INNA method for the sub-area HB logistic regression model. It is an extension of the INNA method in [3]. INNA method is not required to find the posterior modes. Due to the large amount of sub-areas, it would be time-consuming to find all posterior modes, which is why we did not choose the popular INLA method. In detail, we discuss the approximation of the joint posterior density (3).

Notice that the joint posterior density (3) is very complicated and it is the expit part, $\prod_{i=1}^{\ell}\prod_{j=1}^{n_i}\prod_{k=1}^{m_{ij}}\left[\frac{e^{(x'_{ijk}\underset{\sim}{\beta}_{(0)}+w_{ij})y_{ijk}}}{1+e^{x'_{ijk}\underset{\sim}{\beta}_{(0)}+w_{ij}}}\right]$, that causes the difficulties. In the following, we discuss how to

approximate this term to normal density functions by using Laplace approximation, the second-order multivariate Taylor-series approximation and the empirical logistic transform (ELT). This is the key contribution in the paper. Then we use the multiplication rule to approximate the joint posterior density,

$$\pi_a(\underset{\sim}{w}, \underset{\sim}{v}, \underset{\sim}{\beta}, \sigma^2, \delta^2 \mid \underset{\sim}{y}) \propto \pi_a(\underset{\sim}{w} \mid \underset{\sim}{v}, \underset{\sim}{\beta}_{(0)}, \sigma^2, \underset{\sim}{y})\pi_a(\underset{\sim}{v} \mid \beta_0, \delta^2, \underset{\sim}{y})\pi_a(\underset{\sim}{\beta}_{(0)} \mid \underset{\sim}{y})\pi_a(\underset{\sim}{\beta}, \sigma^2, \delta^2 \mid \underset{\sim}{y}),$$

where the first three densities on the right-hand side are all multivariate normal densities. Therefore, we can draw samples and make inference through the approximate joint posterior density.

Let $f(\underset{\sim}{\tau}) = e^{h(\underset{\sim}{\tau})}$ denote the density of a vector of parameters $\underset{\sim}{\tau}$. Let $\underset{\sim}{g}$ denote the gradient vector and $H$ the Hessian matrix at some point $\underset{\sim}{\tau}^*$.

**Lemma 1.** *Let $h(\underset{\sim}{\tau})$ be a logconcave density function with the parameter $\underset{\sim}{\tau}$. Then, $\underset{\sim}{\tau}$ approximately has a multivariate normal distribution,*

$$\underset{\sim}{\tau} \sim Normal(\underset{\sim}{\tau}^* - H^{-1}\underset{\sim}{g}, -H^{-1}).$$

**Proof.** Simply applying the second-order multivariate Taylor series of $h(\underset{\sim}{\tau})$ at $\underset{\sim}{\tau}^*$ gives

$$f(\underset{\sim}{\tau}) \approx f(\underset{\sim}{\tau}^*) + (\underset{\sim}{\tau} - \underset{\sim}{\tau}^*)'\underset{\sim}{g} + \frac{1}{2}(\underset{\sim}{\tau} - \underset{\sim}{\tau}^*)'H(\underset{\sim}{\tau} - \underset{\sim}{\tau}^*).$$

Due to the logconcavity of $h(\underset{\sim}{\tau})$, its Hessian Matrix $-H$ is posit-definite, which can be the covariance matrix. Notice that we are not required to use the mode of $h(\underset{\sim}{\tau})$. We do not need to find the solution of the gradient vector $\underset{\sim}{g} = 0$. Therefore, $\underset{\sim}{\tau}^*$ does not have to be the solution but some other point. It is worth noticing that the term, $-H^{-1}\underset{\sim}{g}$, is a correction to $\underset{\sim}{\tau}^*$. □

To illustrate the approximation steps, we start with a simpler model with flat priors for $\underset{\sim}{\beta}_{(0)}$ and the $\underset{\sim}{w}$, according to model (2). That is,

$$y_{ijk}|w_{ij}, \underset{\sim}{\beta}_{(0)} \overset{ind}{\sim} \text{Bernoulli}\left\{\frac{e^{x'_{\sim}\underset{\sim}{\beta}_{(0)}+w_{ij}}}{1+e^{x'_{ijk}\underset{\sim}{\beta}_{(0)}+w_{ij}}}\right\}, k = 1, \ldots, m_{ij}, j = 1, \ldots, n_i, i = 1, \ldots, \ell,$$

$$p(\underset{\sim}{w}, \underset{\sim}{\beta}_{(0)}) = 1. \tag{4}$$

The joint posterior density is

$$\pi(\underset{\sim}{w}, \underset{\sim}{\beta}_{(0)}|\underset{\sim}{y}) \propto \prod_{i=1}^{\ell}\prod_{j=1}^{n_i}\prod_{k=1}^{m_{ij}}\left\{\frac{e^{(x'_{ijk}\underset{\sim}{\beta}_{(0)}+w_{ij})y_{ijk}}}{1+e^{x'_{ijk}\underset{\sim}{\beta}_{(0)}+w_{ij}}}\right\}. \tag{5}$$

The logarithm of the joint posterior density (or log likelihood) is

$$\Delta = h(\underset{\sim}{\tau}) = \sum_{i=1}^{\ell}\sum_{j=1}^{n_i}\sum_{k=1}^{m_{ij}}\left\{(x'_{ijk}\underset{\sim}{\beta}_{(0)}+w_{ij})y_{ijk} - \log(1+e^{x'_{ijk}\underset{\sim}{\beta}_{(0)}+w_{ij}})\right\}.$$

Let $\underset{\sim}{\tau}' = (\mu', \underset{\sim}{\beta}'_{(0)})$. In our method, we find a convenient point to expand the log-likelihood in a second-order multivariate Taylor-series expansion.

To begin with, let $\bar{y}_{ij} = \frac{1}{m_{ij}}\sum_{k=1}^{m_{ij}} y_{ijk}$. We use the empirical logistic transform $z_{ij}$ to get an estimate of $w_{ij}$, where

$$\hat{w}^*_{ij} = z_{ij} = \log\left\{\frac{\bar{y}_{ij} + \frac{1}{2m_{ij}}}{1 - \bar{y}_{ij} + \frac{1}{2m_{ij}}}\right\}, i = 1, \ldots, \ell; j = 1, \ldots, n_i.$$

First, we discuss how to find the quasi mode of $\underset{\sim}{\beta}_{(0)}$. We plug $\hat{w}_{ij}^*$ into the log likelihood function $\Delta$ and consider it as a function of $\underset{\sim}{\beta}_{(0)}$ only as $q(\underset{\sim}{\beta}_{(0)})$, and we get

$$q(\underset{\sim}{\beta}_{(0)}) = \sum_{i=1}^{\ell} \sum_{j=1}^{n_i} \sum_{k=1}^{m_{ij}} \left[ (\underset{\sim}{x}'_{ijk}\underset{\sim}{\beta}_{(0)} + \hat{w}_{ij}^*)y_{ijk} - \log(1 + e^{\underset{\sim}{x}'_{ijk}\underset{\sim}{\beta}_{(0)} + \hat{w}_{ij}^*}) \right].$$

The first derivative of $q(\underset{\sim}{\beta}_{(0)})$ is

$$q'(\underset{\sim}{\beta}_{(0)}) = \sum_{i=1}^{\ell} \sum_{j=1}^{n_i} \sum_{k=1}^{m_{ij}} \left\{ \underset{\sim}{x}_{ijk}y_{ijk} - \frac{\underset{\sim}{x}_{ijk}e^{(\underset{\sim}{x}'_{ijk}\underset{\sim}{\beta}_{(0)} + \hat{w}_{ij}^*)}}{1 + e^{\underset{\sim}{x}'_{ijk}\underset{\sim}{\beta}_{(0)} + \hat{w}_{ij}^*}} \right\}$$

$$= \sum_{i=1}^{\ell} \sum_{j=1}^{n_i} \sum_{k=1}^{m_{ij}} \left\{ \underset{\sim}{x}_{ijk}y_{ijk} - \underset{\sim}{x}_{ijk}\left[1 + e^{-(\underset{\sim}{x}'_{ijk}\underset{\sim}{\beta}_{(0)} + \hat{w}_{ij}^*)}\right]^{-1} \right\}.$$

Usually we should set $q'(\underset{\sim}{\beta}_{(0)})$ equal to zero and find the modes as the maximum likelihood estimator (MLE) of $\tilde{\beta}_{(0)}$. But here, it is not easy to solve the equation due to the complexity of $q'(\underset{\sim}{\beta}_{(0)})$. We use the first-order Taylor series to approximate it and then simplify $q'(\underset{\sim}{\beta}_{(0)})$ so that we can get quasi-modes of $\underset{\sim}{\beta}_{(0)}$.

The first-order Taylor expansion of $(1 + e^{\underset{\sim}{x}'_{ijk}\tilde{\beta}_{(0)} + \hat{w}_{ij}^*})^{-1}$ equals $(1 - e^{-(\underset{\sim}{x}'_{ijk}\underset{\sim}{\beta}_{(0)} + \hat{w}_{ij}^*)})$. Notice that by Taylor series, $e^{-(\underset{\sim}{x}'_{ijk}\underset{\sim}{\beta}_{(0)} + \hat{w}_{ij}^*)} \approx 1 - (\underset{\sim}{x}'_{ijk}\underset{\sim}{\beta}_{(0)} + \hat{w}_{ij}^*)$. Then we can get

$$q'(\underset{\sim}{\beta}_{(0)}) \approx \sum_{i=1}^{\ell} \sum_{j=1}^{n_i} \sum_{k=1}^{m_{ij}} \left\{ \underset{\sim}{x}_{ijk}y_{ijk} - \underset{\sim}{x}_{ijk}\left[(1 - e^{\underset{\sim}{x}'_{ijk}\underset{\sim}{\beta}_{(0)} + \hat{w}_{ij}^*})\right] \right\}$$

$$\approx \sum_{i=1}^{\ell} \sum_{j=1}^{n_i} \sum_{k=1}^{m_{ij}} \left\{ \underset{\sim}{x}_{ijk}y_{ijk} - \underset{\sim}{x}_{ijk}\left[(1 - 1 + (\underset{\sim}{x}'_{ijk}\underset{\sim}{\beta}_{(0)} + \hat{w}_{ij}^*))\right] \right\}$$

$$= \sum_{i=1}^{\ell} \sum_{j=1}^{n_i} \sum_{k=1}^{m_{ij}} \left\{ \underset{\sim}{x}_{ijk}(y_{ijk} - \hat{w}_{ij}^*) - \underset{\sim}{x}_{ijk}\underset{\sim}{x}'_{ijk}\underset{\sim}{\beta}_{(0)} \right\}.$$

We can get the quasi-modes of $\underset{\sim}{\beta}_{(0)}$ by solving the equation $q'(\underset{\sim}{\beta}_{(0)}) = 0$. That is,

$$\underset{\sim}{\beta}_{(0)}^* = \left[\sum_{i=1}^{\ell} \sum_{j=1}^{n_i} \sum_{k=1}^{m_{ij}} \underset{\sim}{x}_{ijk}\underset{\sim}{x}'_{ijk}\right]^{-1}\left[\sum_{i=1}^{\ell} \sum_{j=1}^{n_i} \sum_{k=1}^{m_{ij}} \underset{\sim}{x}_{ijk}(y_{ijk} - \hat{w}_{ij}^*)\right].$$

Second, we obtain quasi-modes for the $w_{ij}$, a refinement of the $z_i$. Plug $\underset{\sim}{\beta}_{(0)}^*$ into the likelihood function $\Delta$ and consider it as function $w_{ij}$ only:

$$g(w_{ij}) = \sum_{k=1}^{m_{ij}} \left[ (\underset{\sim}{x}'_{ijk}\underset{\sim}{\beta}_{(0)}^* + w_{ij})y_{ijk} - \log(1 + e^{\underset{\sim}{x}'_{ijk}\underset{\sim}{\beta}_{(0)*} + w_{ij}}) \right].$$

Similarly, after applying Taylor expansion, we get the approximate first derivative of $g(w_{ij})$

$$g'(w_{ij}) = \sum_{k=1}^{m_{ij}} \left\{ y_{ijk} - \left[1 + e^{-(\underset{\sim}{x}'_{ijk}\underset{\sim}{\beta}_{(0)}^* + w_{ij})}\right]^{-1} \right\}$$

$$\approx \sum_{k=1}^{m_{ij}} \left\{ y_{ijk} - (1 - e^{-w_{ij}}e^{-\underset{\sim}{x}'_{ijk}\underset{\sim}{\beta}_{(0)}^*}) \right\}.$$

We can obtain the approximate posterior mode of $w_{ij}$ by solving the equation $g'(w_{ij}) = 0$.

$$w_{ij}^* = \log\left\{ \frac{\sum_{k=1}^{m_{ij}} e^{-\underline{x}_{ijk}'\underline{\beta}_{(0)}^*}}{m_{ij}(1 - \bar{y}_{ij})} \right\}.$$

Notice that the term $1 - \bar{y}_{ij}$ in denominator may cause trouble if $\bar{y}_{ij} = 1$ for some $i$s and $j$s. Here, we borrow the idea from ELT and make a small adjustment in order to avoid a zero denominator. That is,

$$w_{ij}^* \approx \log\left\{ \frac{\sum_{k=1}^{m_{ij}} e^{-\underline{x}_{ijk}'\underline{\beta}_{(0)}^*}}{m_{ij}(1 - \bar{y}_{ij} + \frac{1}{2m_{ij}})} \right\} i = 1, \ldots, \ell, \; j = 1, \ldots, n_i.$$

Let $\underline{\tau}^{*\prime} = (\underline{\mu}^{*\prime}, \underline{\beta}_{(0)}^{*\prime})$. Next, we evaluate $\underline{g}$ and H at the quasi-modes $\underline{\tau} = \underline{\tau}^*$ can also be obtained as

$$\underline{g} = \left( \begin{array}{cccc} \frac{\partial\Delta}{\partial w_{11}} & \cdots & \frac{\partial\Delta}{\partial w_{\ell n_\ell}} & \frac{\partial\Delta}{\partial\underline{\beta}_{(0)}} \end{array} \right)^T_{\underline{w}=\underline{w}^*, \, \underline{\beta}_{(0)}=\underline{\beta}_{(0)}^*},$$

$$H = \left( \begin{array}{cccc} \frac{\partial^2\Delta}{\partial w_{11}^2} & \cdots & \frac{\partial^2\Delta}{\partial w_{11}\partial w_{\ell n_\ell}} & \frac{\partial^2\Delta}{\partial w_{11}\partial\underline{\beta}_{(0)}} \\ \vdots & \vdots & \ddots & \vdots \\ 0 & \cdots & \frac{\partial^2\Delta}{\partial w_{\ell n_\ell}^2} & \frac{\partial^2\Delta}{\partial w_{\ell n_\ell}\partial\underline{\beta}_{(0)}} \\ \frac{\partial^2\Delta}{\partial w_{11}\partial\underline{\beta}_{(0)}} & \cdots & \frac{\partial^2\Delta}{\partial w_{\ell n_\ell}\partial\underline{\beta}_{(0)}} & \frac{\partial^2\Delta}{\partial\underline{\beta}_{(0)}^2} \end{array} \right)_{\underline{w}=\underline{w}^*, \underline{\beta}_{(0)}=\underline{\beta}_{(0)}^*}.$$

The partial derivatives can be expressed in terms of response $y_{ijk}$ and covariates $\underline{x}_{ijk}$ as

$$\frac{\partial\Delta}{\partial\underline{\beta}_{(0)}} = \sum_{i=1}^{\ell}\sum_{j=1}^{n_i}\sum_{k=1}^{m_{ij}}\left\{ \underline{x}_{ijk}y_{ijk} - \frac{\underline{x}_{ijk}e^{\underline{x}_{ijk}'\underline{\beta}_{(0)}^*+w_{ij}^*}}{1 + e^{\underline{x}_{ijk}'\underline{\beta}_{(0)}^*+w_{ij}^*}} \right\},$$

$$\frac{\partial\Delta}{\partial w_{ij}} = \sum_{k=1}^{m_{ij}}\left( y_{ijk} - \frac{e^{\underline{x}_{ijk}'\underline{\beta}_{(0)}^*+w_{ij}^*}}{1 + e^{\underline{x}_{ijk}'\underline{\beta}_{(0)}^*+w_{ij}^*}} \right),$$

$$\frac{\partial^2\Delta}{\partial\underline{\beta}_{(0)}^2} = -\sum_{i=1}^{\ell}\sum_{j=1}^{n_i}\sum_{k=1}^{m_{ij}}\frac{\underline{x}_{ijk}\underline{x}_{ijk}'e^{\underline{x}_{ijk}'\underline{\beta}_{(0)}^*+w_{ij}^*}}{(1 + e^{\underline{x}_{ij}'\underline{\beta}_{(0)}^*+w_{ij}^*})^2},$$

$$\frac{\partial^2\Delta}{\partial w_{ij}^2} = -\sum_{k=1}^{m_{ij}}\frac{e^{\underline{x}_{ijk}'\underline{\beta}_{(0)}^*+w_{ij}^*}}{(1 + e^{\underline{x}_{ijk}'\underline{\beta}_{(0)}^*+w_{ij}^*})^2},$$

$$\frac{\partial^2\Delta}{\partial\mu_i\partial\underline{\beta}_{(0)}} = -\sum_{k=1}^{m_{ij}}\frac{\underline{x}_{ijk}e^{\underline{x}_{ijk}'\underline{\beta}_{(0)}^*+w_{ij}^*}}{(1 + e^{\underline{x}_{ijk}'\underline{\beta}_{(0)}^*+w_{ij}^*})^2},$$

where $i = 1, \ldots, \ell, \; j = 1, \ldots, n_i$.

For the convenience of computation, denote $\underline{g} = \left( \begin{array}{c} \underline{g}_1 \\ \underline{g}_2 \end{array} \right)$ and $H = -\left( \begin{array}{cc} D & C' \\ C & B \end{array} \right)$, where

$$\underline{g}_1 = \left( \begin{array}{ccc} \frac{\partial\Delta}{\partial w_{11}} & \cdots & \frac{\partial\Delta}{\partial w_{\ell n_\ell}} \end{array} \right)^T, \underline{g}_2 = \frac{\partial\Delta}{\partial\underline{\beta}_{(0)}},$$

$$B = -\frac{\partial^2\Delta}{\partial\underline{\beta}_{(0)}^2}, C = -\left( \begin{array}{ccc} \frac{\partial^2\Delta}{\partial w_{11}\partial\underline{\beta}_{(0)}} & \cdots & \frac{\partial^2\Delta}{\partial w_{\ell n_\ell}\partial\underline{\beta}_{(0)}} \end{array} \right), D = -\left( \begin{array}{ccc} \frac{\partial^2\Delta}{\partial w_{11}^2} & \cdots & 0 \\ \vdots & \ddots & \vdots \\ 0 & \cdots & \frac{\partial^2\Delta}{\partial w_{\ell n_\ell}^2} \end{array} \right).$$

$$\text{Let} -H^{-1} = \begin{pmatrix} D & C' \\ C & B \end{pmatrix}^{-1} = \begin{pmatrix} E & F' \\ F & G \end{pmatrix}, \text{where}$$

$$E = D^{-1} + D^{-1}C'(B - CD^{-1}C')^{-1}CD^{-1}, F = -(B - CD^{-1}C')^{-1}CD^{-1}, G = (B - CD^{-1}C')^{-1}.$$

**Lemma 2.** *Assuming that the design matrix is full-rank and* $0 < \sum_{k=1}^{m_{ij}} y_{ijk} < m_{ij}$, $j = 1, \ldots, n_i$; $i = 1, \ldots, \ell$, *the posterior density,* $\underset{\sim}{\tau}|y$ *in (5), is logconcave.*

**Proof.** If $0 < \sum_{k=1}^{m_{ij}} y_{ijk} < m_{ij}, i = 1, \ldots, \ell, j = 1, \ldots, n_i$, there are solutions to the gradient vector set to zero.

Let $p_{ijk} = \dfrac{e^{x'_{ijk}\underset{\sim}{\beta}_{(0)} + w_{ij}}}{1 + e^{x'_{ijk}\underset{\sim}{\beta}_{(0)} + w_{ij}}}, k = 1, \ldots, m_{ij}, j = 1, \ldots, n_i, i = 1, \ldots, \ell$. Then, $A$, $B$ and $C$ of the negative Hessian matrix can be written as,

$$B = -\frac{\partial^2 \Delta}{\partial \underset{\sim}{\beta}_{(0)}^2} = \sum_{i=1}^{\ell} \sum_{j=1}^{n_i} \sum_{k=1}^{m_{ij}} p_{ijk}(1 - p_{ijk}) \underset{\sim}{x}_{ijk} \underset{\sim}{x}'_{ijk},$$

$$D = \text{diagonal}(d_{ij}), \quad d_{ij} = \frac{\partial^2 \Delta}{\partial w_{ij}^2} = \sum_{k=1}^{m_{ij}} p_{ijk}(1 - p_{ijk}),$$

$$C = (\underset{\sim}{c}_{ij}), \quad \underset{\sim}{c}_{ij} = \frac{\partial^2 \Delta}{\partial w_{ij} \partial \underset{\sim}{\beta}_{(0)}} = \sum_{k=1}^{m_{ij}} p_{ijk}(1 - p_{ijk}) \underset{\sim}{x}_{ijk},$$

where $j = 1, \ldots, n_i, i = 1, \ldots, \ell$.

It is obvious that $D$ is positive-definite. Thus, to show that $-H$ is positive-definite, we need to show that its Schur complement of $D$, $S = B - CD^{-1}C'$, is positive-definite (e.g., see [35]). Let $\omega_{ijk} = p_{ijk}(1 - p_{ijk}) / \sum_{k=1}^{m_{ij}} p_{ijk}(1 - p_{ijk}), k = 1, \ldots, m_{ij}, j = 1, \ldots, n_i, i = 1, \ldots, \ell$. The Schur complement is

$$S = \sum_{i=1}^{\ell} \sum_{j=1}^{n_i} \sum_{k=1}^{m_{ij}} p_{ijk}(1 - p_{ijk}) \sum_{k=1}^{m_{ij}} \omega_{ijk} \underset{\sim}{x}_{ijk} \underset{\sim}{x}'_{ijk} - \sum_{i=1}^{\ell} \sum_{j=1}^{n_i} \sum_{k=1}^{m_{ij}} p_{ijk}(1 - p_{ijk}) \sum_{k=1}^{m_{ij}} \omega_{ijk} \underset{\sim}{x}_{ijk} \sum_{k=1}^{m_{ij}} \omega_{ijk} \underset{\sim}{x}'_{ijk}.$$

It is now easy to show that

$$S = \sum_{i=1}^{\ell} \sum_{j=1}^{n_i} \sum_{k=1}^{m_{ij}} \omega_{ijk} \left( \underset{\sim}{x}_{ijk} - \sum_{k=1}^{m_{ij}} \omega_{ijk} \underset{\sim}{x}_{ijk} \right) \left( \underset{\sim}{x}_{ijk} - \sum_{k=1}^{m_{ij}} \omega_{ijk} \underset{\sim}{x}_{ijk} \right)'.$$

Therefore, $-H$ is positive-definite, and $\underset{\sim}{\tau}|y$ is logconcave. □

Finally, according to the Lemmas 1 and 2, we can establish the approximation Theorem.

**Theorem 1.** *Assuming that the design matrix is full-rank and* $0 < \sum_{k=1}^{m_{ij}} y_{ijk} < m_{ij}$, $j = 1, \ldots, n_i$, $i = 1, \ldots, \ell$, *the posterior density,* $\underset{\sim}{\tau}|y$ *in (5) is approximately a multivariate normal density, and the conditional posterior density of* $\underset{\sim}{w}|\underset{\sim}{\beta}_{(0)}, y$ *and* $\underset{\sim}{\beta}_{(0)}|y$ *can also be approximated by multivariate normal distributions.*

**Proof.** The proof is given in Appendix B. □

Therefore, we can approximate that logit expit term $\prod_{i=1}^{\ell} \prod_{j=1}^{n_i} \prod_{k=1}^{m_{ij}} \left[ \dfrac{e^{(x'_{ijk}\underset{\sim}{\beta}_{(0)} + w_{ij})y_{ijk}}}{1 + e^{x'_{ijk}\underset{\sim}{\beta}_{(0)} + w_{ij}}} \right]$ into two multivariate densities by Theorem 1. And then we can get our approximate two-fold Bayesian logistic regression model.

Recall the posterior density of our two-fold logistic model is

$$\pi(\underset{\sim}{w}, \underset{\sim}{v}, \underset{\sim}{\beta}, \sigma^2, \delta^2 \mid y) \propto \pi(y|\underset{\sim}{w}, \underset{\sim}{\beta}_{(0)}) \pi(\underset{\sim}{w} \mid \underset{\sim}{v}, \sigma^2) \pi(\underset{\sim}{v} \mid \beta_0, \delta^2) \pi(\underset{\sim}{\beta}_{(0)}, \beta_0, \sigma^2, \delta^2)$$

The likelihood function $\pi(y|w, \beta_{(0)})$ can be approximated by the multivariate normal distribution by Theorem 1. Combining the prior values of $w$ and $v$ given by our Bayesian Logistic model and the results in Theroem 1, we can obtain our INNA model

$$w|\beta_{(0)}, y \sim \text{Normal}\{\mu_w - D^{-1}C'(\beta_{(0)} - \mu_\beta), D^{-1}\}$$

$$\beta_{(0)}|y \sim \text{Normal}\{\mu_\beta, G\}$$

$$w|v, \sigma^2 \stackrel{ind}{\sim} \text{Normal}(\mu_v, \sigma^2 I),$$

$$v|\beta_0, \delta^2 \stackrel{iid}{\sim} \text{Normal}(\beta_0 j, \delta^2 I),$$

$$\pi(\beta_{(0)}, \beta_0, \delta^2, \sigma^2) \propto \frac{1}{(1+\delta^2)^2} \frac{1}{(1+\sigma^2)^2}, \delta^2 > 0, \sigma^2 > 0,$$

where $\mu'_v = (\underbrace{v_1, \ldots, v_1}_{n_1} \cdots \underbrace{v_\ell, \ldots, v_\ell}_{n_\ell})'$ and $j$ is a vector of ones.

Using Bayes' Theorem and the multiplication rule, the posterior density $\pi(w, v, \beta, \sigma^2, \delta^2 \mid y)$ can be approximated as

$$\pi_a(w, v, \beta, \sigma^2, \delta^2 \mid y) \propto \pi_a(w \mid v, \beta_{(0)}, \sigma^2, y)\pi_a(v \mid \beta_0, \delta^2, y)\pi_a(\beta_{(0)} \mid y)\pi_a(\beta, \sigma^2, \delta^2 \mid y)$$

$$= e^{-\frac{1}{2}\left\{\left[w - \left(\mu_w - D^{-1}C'(\beta_{(0)} - \mu_\beta)\right)\right]'D\left[w - \left(\mu_w - D^{-1}C'(\beta_{(0)} - \mu_\beta)\right)\right]\right\}}$$

$$\times e^{-\frac{1}{2}\left\{\left[w - \mu_v\right]'(\sigma^2 I)^{-1}\left[w - \mu_v\right] + \left[v - \beta_0 j\right]'(\delta^2 I)^{-1}\left[v - \beta_0 j\right] + \left[\beta_{(0)} - \mu_\beta\right]'G^{-1}\left[\beta_{(0)} - \mu_\beta\right]\right\}}$$

$$\times \frac{|D|^{1/2}}{|\delta^2 I|^{1/2}|\sigma^2 I|^{1/2}|G|^{1/2}} \frac{1}{(1+\sigma^2)^2} \frac{1}{(1+\delta^2)^2} \tag{6}$$

Therefore, we can get the following key result.

**Theorem 2.** *Using the multiplication rule, the joint posterior density, $\pi(w, v, \beta, \sigma^2, \delta^2 \mid y)$ in (6), can be approximated by*

$$\pi_a(w, v, \beta, \sigma^2, \delta^2 \mid y) \propto \pi_a(w \mid v, \beta_{(0)}, \sigma^2, y)\pi_a(v \mid \beta_0, \delta^2, y)\pi_a(\beta_{(0)} \mid y)\pi_a(\beta, \sigma^2, \delta^2 \mid y),$$

*where the first three densities on the right-hand side are all multivariate normal densities.*

**Proof.** The proof is given in Appendix C. □

The INNA is actually a random sampler. First, we draw samples for $\sigma^2, \delta^2$ from $\pi(\sigma^2, \delta^2|y)$. The posterior distribution of $\sigma^2, \delta^2|y$ does not have standardized form. Here, we use the grid method and numerical integration to sample $\sigma^2$ and $\delta^2$. Since $0 < \sigma^2 < \infty$ and $0 < \delta^2 < \infty$, we make a transformation to $\phi_1 = \frac{1}{1+\sigma^2}$ and $\phi_2 = \frac{1}{1+\delta^2}$ so that we get $0 < \phi_1 < 1$ and $0 < \phi_2 < 1$. Then, the posterior density of $\phi_1, \phi_2|y$ is

$$\pi_a(\phi_1, \phi_2|y) \propto \left\{ \left| \begin{array}{cc} \delta_0^2 & \gamma' \\ \gamma & \Delta_{(0)} \end{array} \right|^{-\frac{1}{2}} \times \prod_{i=1}^{l} \left( \frac{1}{\sum_{j=1}^{n_i} \sigma_{ij}^2} + \frac{1}{\delta^2} \right)^{\frac{1}{2}} \frac{1}{|\delta^2 D + \frac{\delta^2}{\sigma^2}I|^{1/2}} \right\}_{\phi_1 = \frac{1}{1+\sigma^2}, \phi_2 = \frac{1}{1+\delta^2}}$$

$$\times \exp\left\{ -\frac{1}{2}(\mu_w + D^{-1}C'\mu_\beta)'(D^{-1} + \sigma^2 I + \delta^2 I)^{-1}(\mu_w + D^{-1}C'\mu_\beta) + \mu'_\beta G^{-1}\mu_\beta \right\}$$

$$\times \exp\left\{ -\frac{1}{2} \left( \begin{array}{c} \beta_0 - \omega_0 \\ \beta_{(0)} - \omega_{(0)} \end{array} \right)' \left( \begin{array}{cc} \delta_0^2 & \gamma' \\ \gamma & \Delta_{(0)} \end{array} \right) \left( \begin{array}{c} \beta_0 - \omega_0 \\ \beta_{(0)} - \omega_{(0)} \end{array} \right) \right\}_{\phi_1 = \frac{1}{1+\sigma^2}, \phi_2 = \frac{1}{1+\delta^2}}.$$

We need to draw $\phi_1, \phi_2$ together. The joint density can be rewritten as

$$\pi(\phi_1, \phi_2 | \underset{\sim}{y}) = \pi(\phi_2 | \phi_1) \pi(\phi_1 | \underset{\sim}{y}) = \pi(\phi_2 | \phi_1) \int_0^1 \pi(\phi_1, \phi_2 | \underset{\sim}{y}) d\phi_2.$$

We plug each grid of $\phi_1 \in (0, 1)$ into $\int_0^1 \pi(\phi_1, \phi_2 | \underset{\sim}{y}) d\phi_2$ and then use numerical integration to get the density of $(\phi_1 | \underset{\sim}{y})$. After we plug all the 100 grids, we can get 100 value of $\pi(\phi_1 | \underset{\sim}{y})$ and then draw $\phi_1$ from them, i.e. $\phi_1^{(h)}$. Next, we plug $\phi_1^{(h)}$ into $\pi(\phi_2 | \phi_1)$ and use grid method to draw $\phi_2^{(h)}$. We repeat those steps 10,000 times to get the sample of $(\phi_1^{(h)}, \phi_2^{(h)}), h = 1, \dots, 10,000$. Once we get samples for $\phi_1, \phi_2$, we transform them back to $\sigma^2$ and $\delta^2$ respectively. Second, given $\sigma^2, \delta^2$, we can simply draw samples of $\beta$ from the approximate multivariate normal distribution $\pi_a(\underset{\sim}{\beta} \mid \sigma^2, \delta^2, \underset{\sim}{y})$. Third, we can draw samples of $\nu_i$ independently given $\beta, \delta^2$ and data from the approximate normal distribution $\pi_a(\underset{\sim}{\nu} | \beta_0, \delta^2, \underset{\sim}{y})$. Finally, samples of $w_{ij}$ independently given $\underset{\sim}{\nu}, \beta, \sigma^2$ can be obtained from the approximate normal distribution $\pi_a(\underset{\sim}{w} | \underset{\sim}{\nu}, \beta_{(0)}, \sigma^2, \underset{\sim}{y})$. Notice that the last three steps are very simple, just drawing samples from normal densities. In addition, $w_{ij}$ and $\nu_i$ are all independent so that we can draw them simultaneously. Therefore, those latter steps permit fast computing.

In order to check if INNA method can provide resonal results, we apply the MCMC logistic regression exact method to the sub-area model. The idea of exact method is to get full conditional posterior distributions for all of the parameters in the model, and then get a large number of independent samples of each parameter with its full conditional posterior density. Details are given in Appendix A.

There are two differences between these two methods. First, both methods are sampling-based. The approximate method implements random samples and the exact method uses numerical integration method and Markov chains. Second, $\pi_a(\beta, \sigma^2, \delta^2 \mid \underset{\sim}{y})$ is used for the INNA method. In the exact method, a Metroplis step is used for the $\pi(\beta, \sigma^2, \delta^2 \mid \underset{\sim}{y})$. This is very time-consuming in the exact method. On the other hand, the exact method actually uses the INNA method. We use M-H sampler draw samples for $\underset{\sim}{\nu}$ and $\underset{\sim}{w}$, respectively. Proposal functions are $\pi_a(\underset{\sim}{\nu} \mid \beta_{(0)}, \beta_0, \sigma^2, \underset{\sim}{y})$ and $\pi_a(\underset{\sim}{w} \mid \underset{\sim}{\nu}, \beta_{(0)}, \sigma^2, \underset{\sim}{y})$, respectively, from the INNA method.

## 4. Numerical Example

### 4.1. Nepal Living Standards Survey II

The performance of our method is studied using the Nepal Living Standard Survey (NLSS II), conducted in the years 2003–2004. The main objective of the NLSS II is to track changes in and the progress of national living standards and social indicators of the Nepalese population. It is an integrated survey which covers samples from the whole country and runs throughout the year.

The NLSS II gathers information on a variety of aspects. It has collected data on demographics, housing, education, health, fertility, employment, income, agricultural activity, consumption, and various other areas. The sampling design of NLSS II is two-stage stratified sampling. Nepal is stratified into Primary Sampling Units (PSUs) and, within each PSU, there are a number of households (sub-area) selected. All household members in the sample were interviewed.

In detail, the NLSS II has records for 20,263 individuals from 3912 households (sub-areas) from 326 PSUs (areas) from a population of 60,262 households and about two million Nepalese. A sample of PSUs was selected from the strata using probability proportional to size (PPS) sampling and 12 households were systematically selected from each PSU. The survey is self-weighted and some adjustments were made after conducting the survey for non-responses or missing data. For simplicity, in this paper, we assume all samples have the same weight. Table 1 shows the distribution of all samples by stratum.

**Table 1.** Distribution of wards and households in the sample.

| Strata | Mountains | Kathemandu | Urban Hill | Rural Hills | Urban Tarai | Rural Tarai | Total |
|---|---|---|---|---|---|---|---|
| PSU | 32 | 34 | 28 | 96 | 34 | 102 | 326 |
| Households | 384 | 408 | 336 | 1152 | 408 | 1224 | 3912 |
| Individuals | 1949 | 1954 | 1467 | 5755 | 2104 | 7034 | 20,263 |

We chose four relevant covariates which can influence health status from the same NLSS II survey for our two-fold logistic regression model. They are age, nativity, sex and religion. We created binary variables for nativity (Indigenous = 1, Non-indigenous = 0) and religion ((Hindu = 1, Non-Hindu = 0), sex (Male = 1, Female = 0). Table 2 shows the details of these four covariates. In the model fitting, we standardized the age covariate. Older age and a child's age are more vulnerable times than younger age. Indigenous people can have different health statuses from migrated people.

**Table 2.** The descriptives of 4 covariates.

| Covariates | | Frequency | Percentage |
|---|---|---|---|
| Age | 0–14 | 7765 | 38.32 |
| | 15–59 | 10,951 | 54.04 |
| | 60+ | 1547 | 7.64 |
| Gender | Male | 9763 | 48.18 |
| | Female | 10,500 | 51.82 |
| Nativity | Indigeous | 11,903 | 41.25 |
| | Non-Indigous | 8,360 | 58.75 |
| Religion | Hingdu | 16,378 | 80.83 |
| | Non-Hingdu | 3385 | 19.17 |

According to the 2001 census data, only about 0.091% of households and only 0.904% of PSU were sampled. The NLSS II was designed to provide reliable estimates only at the stratum level, or even larger areas than the stratum. It cannot give estimates in small areas (PSU or household level) since the sample sizes are too small. Therefore, we need to use statistical models to fit the available data and find reliable estimates in small areas. In our study, we chose the binary variable, health status, from the health section of the questionnaire.

*4.2. Numerical Comparison*

We used data from NLSS II to illustrate our sub-area logistic regression model. We predicted the household proportions of members in good health for 18,924 households (sampled and non-sampled). Bayesian bootstrap by [36] was applied to get non-sampled auxiliary information. This analysis was based on 1224 sample households from 102 wards (PSUs) in strata 6. Our primary purposes were to show that our model can provide good estimates and to compare the approximate method with the exact method when there are random effects at the household level.

We used Rcpp [37] and RcppArmadillo [38] packages in R [39] to fit the model based on both the approximation, INNA, method and the exact method to this NLSS II dataset. For the INNA method, we began with 10,000 iterations and a burn-in of 1000 and we kept only every ninth sample. Finally, 1000 samples were obtained for constructing the posterior distributions of all the parameters. The exact method was very time-consuming, taking about 30 h to finish. However, the INNA approximation method can get samples in 8 min. When we have a large number of areas or sub-areas, the approximation method will make enormous savings.

Convergence diagnostics were conducted. The convergences of the hyperparameters $(\beta, \sigma^2, \delta^2)$ were monitored by the Geweke test of stationarity [40] and the effective sample sizes. The p-values and effective sample sizes are shown in Table 3, resulting in good

convergence for both methods. Table 3 also shows the posterior means (PMs) and associated posterior standard deviations (PMs) of the hyperparameters. The PMs are very close between these two methods. The PSDs are slightly larger for the exact method than for the INNA method, but they are reasonably close.

**Table 3.** Posterior means (PM), associated posterior standard deviations (PSD), Geweke test *p*-values and effective sample sizes (ESS) of hyperparameters based on the INNA and exact method.

| Method | INNA | | | | Exact | | | |
|---|---|---|---|---|---|---|---|---|
| **Estimator** | **PM** | **PSD** | ***p*-Value** | **ESS** | **PM** | **PSD** | ***p*-Value** | **ESS** |
| $\beta_0$ | 0.802 | 0.323 | 0.688 | 780 | 0.840 | 0.405 | 0.264 | 1000 |
| $\beta_1$ | 0.675 | 0.015 | 0.759 | 857 | 0.654 | 0.016 | 0.156 | 858 |
| $\beta_2$ | 0.302 | 0.017 | 0.453 | 894 | 0.313 | 0.019 | 0.655 | 1000 |
| $\beta_3$ | −0.862 | 0.017 | 0.605 | 915 | −0.824 | 0.017 | 0.418 | 863 |
| $\beta_4$ | −0.338 | 0.013 | 0.718 | 937 | −0.360 | 0.011 | 0.839 | 1000 |
| $\sigma^2$ | 115.248 | 35.601 | 0.408 | 1000 | 108.959 | 38.719 | 0.615 | 911 |
| $\delta^2$ | 30.002 | 1.146 | 0.448 | 731 | 29.034 | 1.697 | 0.490 | 780 |

In Figures 1–3, we compare the PMs, PSDs and posterior coefficient of variations (PCVs) in the household level as our primary purpose. We can see that the PMs are very close, nearly lying on the 45-degree line through the origin. The PSDs are slightly spread out and thicker, but all points still lie on the 45-degree line and so do the PCVs. Overall, these approximations are acceptable in the data analysis. Figures 4–6 we compare respectively to the PMs, PSDs and PCVs at the ward level. The plots of the PMs are still very good. Notice that other two plots of PSDs and PCVs are more spread out than those in the household level. Again, though, the approximate method and the exact method are reasonably close.

We also compare the approximate method with the exact method using the five number summaries (the minimum values, the first quartiles, the median, the third quartiles, and the maximum values) with respect to the PMs, PSDs and PCVs of the finite population proportions at the household level and ward level in Tables 4 and 5. The PMs from both methods at the household level have larger variations than those at the ward level. The PCVs at the ward level are generally much smaller than at the household level. The summaries of the PMs, PSDs and PCVs within households and wards between the approximate and exact methods are very close.

**Table 4.** Comparison of posterior inference about the finite population proportions using the five-number summaries at the household level.

| Households | | **Min** | **Q1** | **Mean** | **Q3** | **Max** |
|---|---|---|---|---|---|---|
| PM | INNA | 0.049 | 0.541 | 0.565 | 0.584 | 0.981 |
| | Exact | 0.050 | 0.542 | 0.566 | 0.584 | 0.984 |
| PSD | INNA | 0.040 | 0.246 | 0.441 | 0.465 | 0.500 |
| | Exact | 0.038 | 0.248 | 0.442 | 0.466 | 0.501 |
| PCV | INNA | 0.041 | 0.379 | 0.762 | 0.852 | 1.549 |
| | Exact | 0.039 | 0.384 | 0.768 | 0.852 | 1.577 |

**Table 5.** Comparison of posterior inference about the finite population proportions using five-number summaries at the ward level.

| Wards | | Min | Q1 | Mean | Q3 | Max |
|---|---|---|---|---|---|---|
| PM | INNA | 0.449 | 0.537 | 0.563 | 0.589 | 0.684 |
| | Exact | 0.450 | 0.537 | 0.564 | 0.590 | 0.683 |
| PSD | INNA | 0.056 | 0.059 | 0.063 | 0.066 | 0.077 |
| | Exact | 0.058 | 0.061 | 0.064 | 0.066 | 0.077 |
| PCV | INNA | 0.095 | 0.103 | 0.113 | 0.121 | 0.163 |
| | Exact | 0.097 | 0.103 | 0.113 | 0.122 | 0.165 |

We conclude that the approximation method at the household level is reasonable. The approximation is desirable because one can perform the computations in real time.

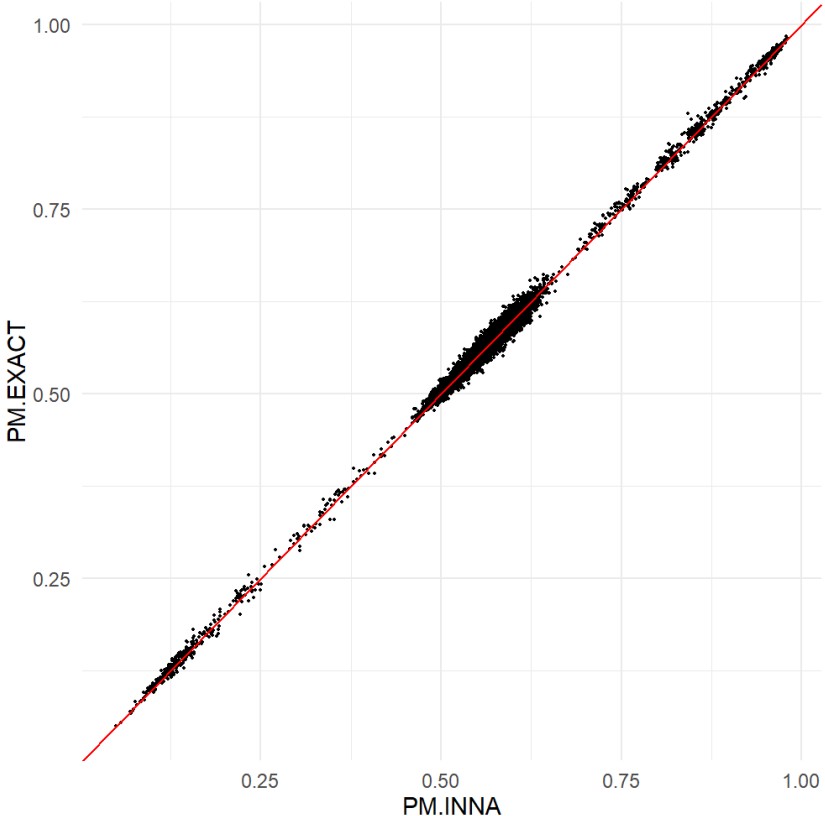

**Figure 1.** Comparison of the INNA method and the exact method using the PSDs of the household proportions.

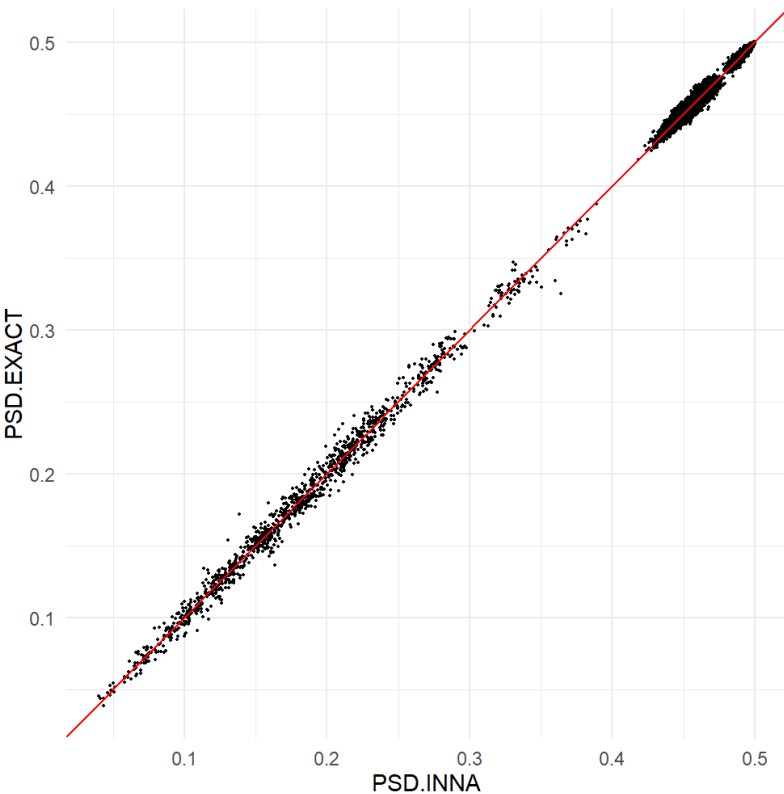

**Figure 2.** Comparison of the INNA method and the exact method using the PSDs of the household proportions.

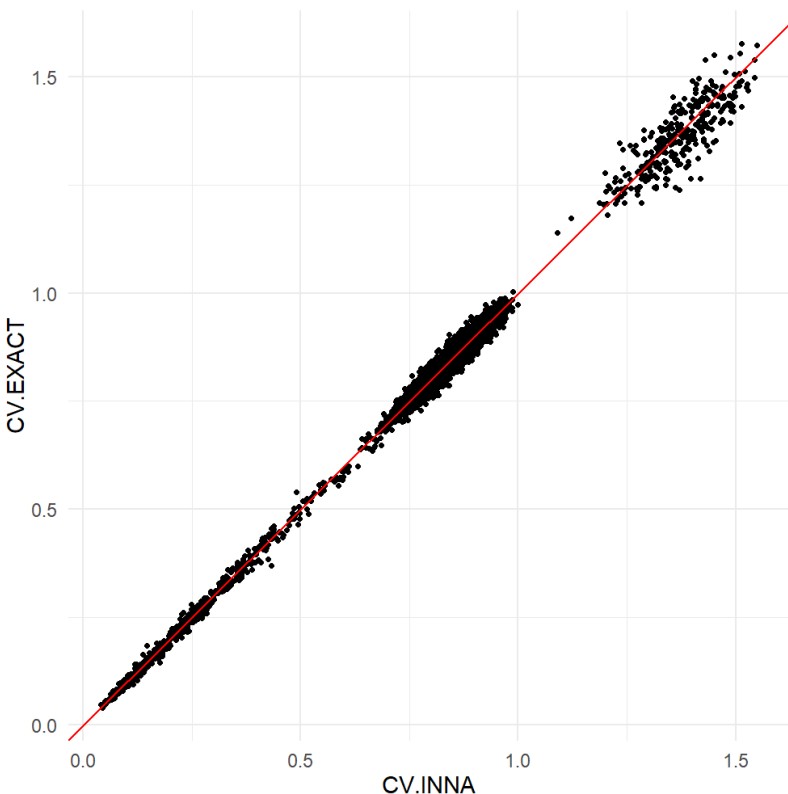

**Figure 3.** Comparison of the INNA method and the exact method using the CVs of the household proportions.

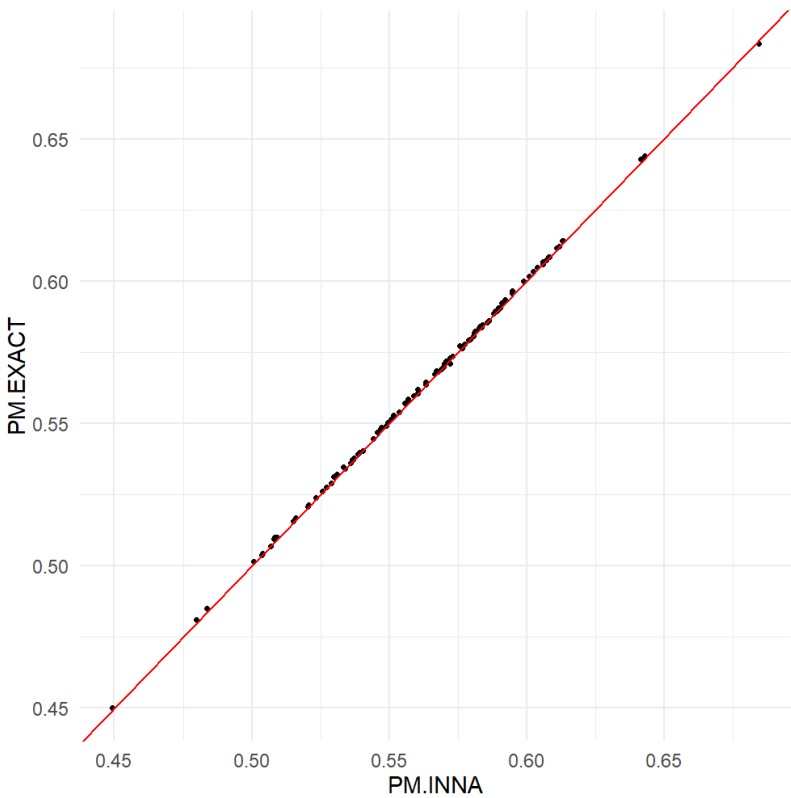

**Figure 4.** Comparison of the INNA method and the exact method using the PMs of the ward proportions.

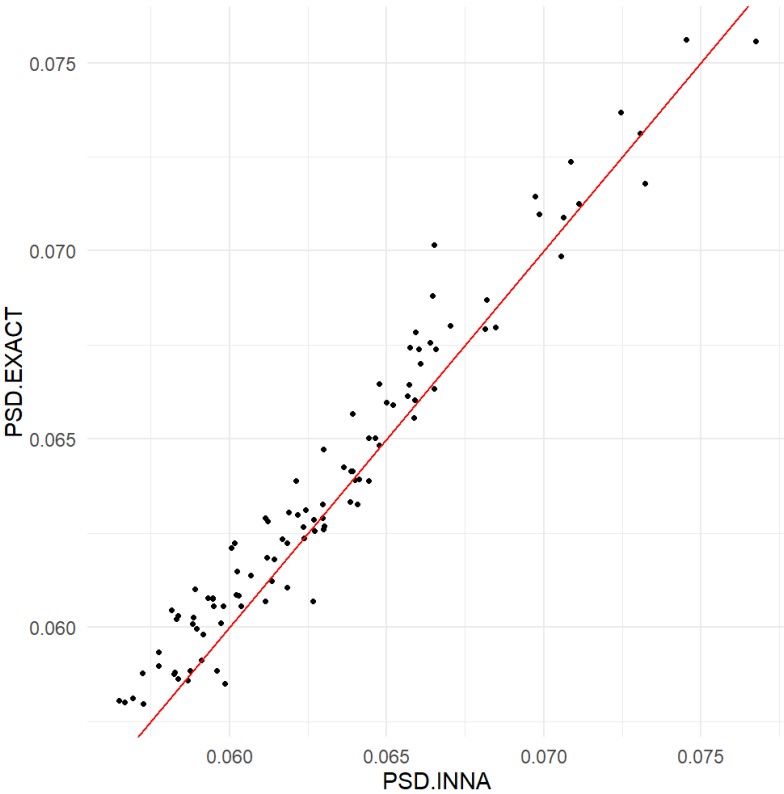

**Figure 5.** Comparison of the INNA method and the exact method using the PSDs of the ward proportions.

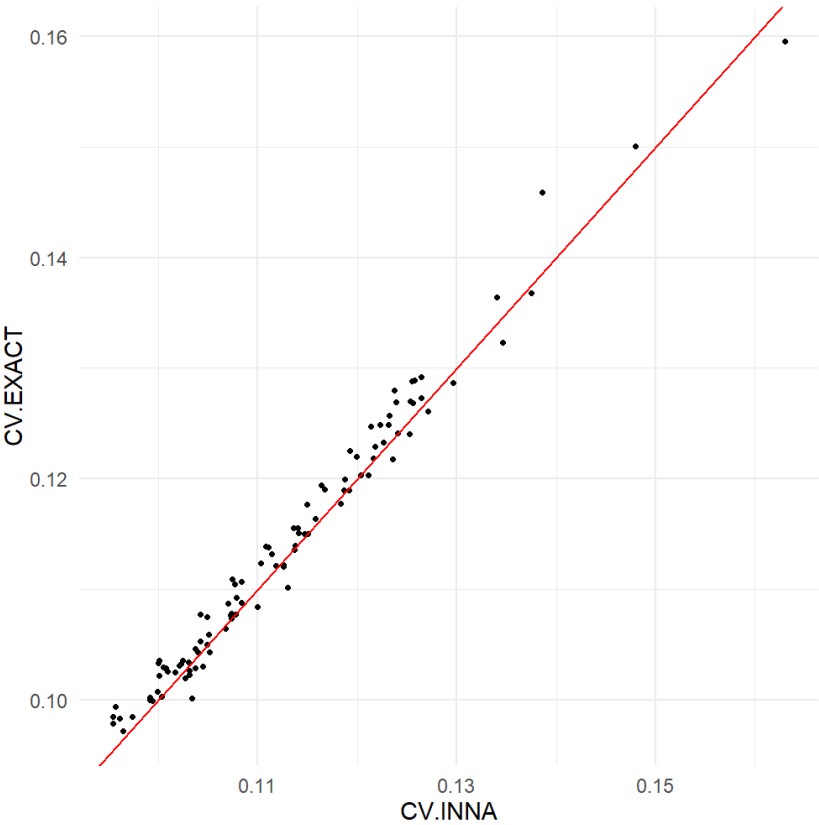

**Figure 6.** Comparison of the INNA method and the exact method using the CVs of the ward proportions.

## 5. Conclusions and Future Work

The sub-area HB logistic regression model can be applied to analyze the binary response variable. This model is an extension of the HB logistic regression area-level model, which ignores the actual hierarchical structure of the data. We propose an approximation method, INNA, to fit the model. For large datasets, it is very unrealistic to use the MCMC method to fit the model. We propose the approximation method, INNA, which saves time significantly because there is no need to compute numerous modes. In the numerical example, we can show that INNA can provide reliable estimates as well. An illustrative example of the NLSS II is presented in order to compare the approximation method and the exact method. It shows that, when there are a large number of areas and sub-areas, the approximation method can be efficient and it can also provide reasonable estimates.

INNA is a method for approximate Bayesian inference based on Laplace's method, the second-order multivariate Taylor-series approximation and the empirical logistic transform (ELT). It can be applied to all HB logistic regression models, for which it can be a fast and accurate alternative to the Markov chain Monte Carlo methods. The comparison and model results illustrate the performance of the INNA methods based on the sub-area model.

There will be many future works on the two-fold small areas model. First, in this paper, we assume equal survey weights since the NLSS II is a self-weighted sampling. However, after the data are collected, the sampling weights are usually adjusted for various characteristics or based on nonresponse as well. Incorporating those survey weights into the model is also very important. Generally, we need to consider these weights in the model. The NLSS II is a national population-based survey. We should rescale the sample weights to sum to an equivalent sample size. That is, we consider the adjusted weight as

$$w^*_{ijk} = \hat{n}\left(\frac{w_{ijk}}{\sum\limits_{i=1}^{\ell}\sum\limits_{j=1}^{n_i}\sum\limits_{k=1}^{m_{ij}} w_{ijk}}\right), \text{ where } \hat{n} = \frac{(\sum\limits_{i=1}^{\ell}\sum\limits_{j=1}^{n_i}\sum\limits_{k=1}^{m_{ij}} w_{ijk})^2}{\sum\limits_{i=1}^{\ell}\sum\limits_{j=1}^{n_i}\sum\limits_{k=1}^{m_{ij}} w_{ijk}^2} \text{ as an equivalent sample. Introducing the}$$

sampling weights, we can obtain an updated normalized likelihood function. Based on the updated likelihood function and the same prior in the two-fold model, we can have a full Bayesian analysis on the updated model and then project the finite population proportion of the family members with good health in each household.

Second, we focus on the binary data. Actually, there are four options in the health status questionnaire. The Multinomial-Dirichlet model can be an extension of the polychotomous data. Third, the two-fold sub-area level models can also be extended to three-fold models if the data have an additional hierarchical structure; actually, the NLSS II has this structure (households within wards, wards within districts). Fourth, in our models, we consider parametric priors. Introducing the Dirichlet process as a prior might make our method more robust to its specifications.

**Author Contributions:** Conceptualization, B.N. and L.C.; methodology, B.N. and L.C.; software, L.C.; validation, B.N. and L.C.; formal analysis, L.C.; investigation, L.C.; resources, B.N.; writing—original draft preparation, L.C.; writing—review and editing, B.N. and L.C.; visualization, L.C.; supervision, B.N. All authors have read and agreed to the published version of the manuscript.

**Funding:** Balgobin Nandram was supported by a grant from the Simons Foundation (#353953, Balgobin Nandram).

**Conflicts of Interest:** The authors declare no conflict of interest.

## Appendix A. Exact Method for Sub-Area Logistic Regression Model

Recall that the joint posterior distribution of our two-fold logistic regression model is the joint posterior density for the parameters is

$$\pi(\underset{\sim}{v}, \underset{\sim}{w}, \underset{\sim}{\beta}, \sigma^2, \delta^2 | \underset{\sim}{y}) \propto \prod_{i=1}^{\ell} \prod_{j=1}^{n_i} \prod_{k=1}^{m_{ij}} \left[ \frac{e^{(x'_{ijk}\underset{\sim}{\beta}_{(0)} + w_{ij})y_{ijk}}}{1 + e^{x'_{ijk}\underset{\sim}{\beta}_{(0)} + w_{ij}}} \right] \times \left( \frac{1}{\sqrt{2\pi\sigma^2}} \right)^n \exp\left\{ -\sum_{i=1}^{l} \sum_{j=1}^{ni} \frac{(w_{ij} - v_i)^2}{2\sigma^2} \right\}$$

$$\times \left( \frac{1}{\sqrt{2\pi\delta^2}} \right)^l \exp\left\{ -\sum_{i=1}^{l} \frac{(v_i - \beta_0)^2}{2\delta^2} \right\} \frac{1}{(1 + \sigma^2)^2} \frac{1}{(1 + \delta^2)^2}.$$

We can see that the form of the joint posterior density is very complicated. It is very time consuming to draw all the posterior samples if applying the exact MCMC method. But the exact method will provide reliable estimates of all parameters, so in order to test the performance of our approximation method, we need to apply MCMC method on our model and then compare the performance of two methods. We use Metropolis-Hastings sampler to draw samples for $\underset{\sim}{\beta}, \sigma^2, \delta^2$ together and then draw $\underset{\sim}{v}$ given $\underset{\sim}{\beta}, \sigma^2, \delta^2$ samples. At last, we use MH method to draw $\underset{\sim}{w}$ given $\underset{\sim}{v}, \underset{\sim}{\beta}, \sigma^2, \delta^2$ samples.

In order to draw samples for $\underset{\sim}{\beta}, \sigma^2, \delta^2$ together, we need to integrate out $\underset{\sim}{w}$ and $\underset{\sim}{v}$. First, we integrate out $\underset{\sim}{w}$ from the joint posterior density $\pi(\underset{\sim}{v}, \underset{\sim}{w}, \underset{\sim}{\beta}, \sigma^2, \delta^2 | \underset{\sim}{y})$ to get

$$\pi(\underset{\sim}{v}, \underset{\sim}{\beta}, \sigma^2, \delta^2 | \underset{\sim}{y}) \propto \int_{\Omega} \prod_{i=1}^{\ell} \left\{ \prod_{j=1}^{n_i} \left[ \prod_{k=1}^{m_{ij}} \frac{e^{(x'_{ijk}\underset{\sim}{\beta}_{(0)} + \mu_i + w_{ij})y_{ij}}}{1 + e^{x'_{ijk}\underset{\sim}{\beta}_{(0)} + \mu_i + w_{ij}}} \right] \frac{1}{\sqrt{2\pi\sigma^2}} e^{-\frac{(w_{ij} - v_i)^2}{2\sigma^2}} \right\} d\underset{\sim}{w}$$

$$\times \left( \frac{1}{\sqrt{2\pi\delta^2}} \right)^l \exp\left\{ -\sum_{i=1}^{l} \frac{(v_i - \beta_0)^2}{2\delta^2} \right\} \frac{1}{(1 + \sigma^2)^2} \frac{1}{(1 + \delta^2)^2}.$$

Notice that the integrant is not any simple distribution function, so we use Monte Carlo numberical integration to approximate the integrals. Let $z_{ij}^w = \frac{w_{ij} - v_i}{\sigma}$. Notice that $z_{ij}^w$ follows standard normal distribution. For standard normal density, 99.7% of data will fall within 3 standard deviations of the mean, which corresponds to the interval $[-3, 3]$. Therefore, we bounded the integration domain to $[-3, 3]$ and divide the interval to M equal

subintervals $[p_{a-1}, p_a], a = 1, \ldots, M$. Then we can get an approximate but very accurate joint density

$$
\pi(\underset{\sim}{\nu}, \underset{\sim}{\beta}, \sigma^2, \delta^2 | \underset{\sim}{y}) \propto \prod_{i=1}^{\ell} \prod_{j=1}^{n_i} \left\{ \sum_{a=1}^{M} \int_{p_{a-1}}^{p_a} \frac{e^{\sum_{k=1}^{m_{ij}} (\underset{\sim}{x}'_{ijk} \underset{\sim}{\beta}_{(0)} + w_{ij}) y_{ijk}}}{\prod_{k=1}^{m_{ij}} \left[ 1 + e^{\underset{\sim}{x}'_{ijk} \underset{\sim}{\beta}_{(0)} + w_{ij}} \right]} \frac{1}{\sqrt{2\pi\sigma^2}} e^{-\frac{(w_{ij} - \nu_i)^2}{2\sigma^2}} dw_{ij} \right\}
$$

$$
\times \left( \frac{1}{\sqrt{2\pi\delta^2}} \right)^l \exp\left\{ -\sum_{i=1}^{l} \frac{(\nu_i - \beta_0)^2}{2\delta^2} \right\} \frac{1}{(1+\sigma^2)^2} \frac{1}{(1+\delta^2)^2}
$$

$$
\propto \prod_{i=1}^{\ell} \prod_{j=1}^{n_i} \left\{ \sum_{a=1}^{M} \int_{p_{a-1}}^{p_a} \frac{e^{\sum_{k=1}^{m_{ij}} (\underset{\sim}{x}'_{ijk} \underset{\sim}{\beta}_{(0)} + \sigma z_{ij}^w + \nu_i) y_{ijk}}}{\prod_{k=1}^{m_{ij}} \left[ 1 + e^{\underset{\sim}{x}'_{ijk} \underset{\sim}{\beta}_{(0)} + \sigma z_{ij}^w + \nu_i} \right]} \frac{1}{\sqrt{2\pi}} e^{-\frac{(z_{ij}^w)^2}{2}} dz_{ij}^w \right\}
$$

$$
\times \left( \frac{1}{\sqrt{2\pi\delta^2}} \right)^l \exp\left\{ -\sum_{i=1}^{l} \frac{(\nu_i - \beta_0)^2}{2\delta^2} \right\} \frac{1}{(1+\sigma^2)^2} \frac{1}{(1+\delta^2)^2}
$$

Let $\bar{z}_a^w = \frac{p_a - p_{a-1}}{2}$, which is the midpoint of each interval $[p_{a-1}, p_a], a = 1, \ldots, M$. We use midpoint rule to approximate the definite integrals. We divide the interval $[-3, 3]$ into 100 subintervals, and so we use 100 midpoints to get the approximate joint posterior distribution

$$
\pi(\underset{\sim}{\nu}, \underset{\sim}{\beta}, \sigma^2, \delta^2 | \underset{\sim}{y}) \approx \prod_{i=1}^{\ell} \prod_{j=1}^{n_i} \left\{ \sum_{a=1}^{100} \frac{e^{\sum_{k=1}^{m_{ij}} (\underset{\sim}{x}'_{ijk} \underset{\sim}{\beta}_{(0)} + \sigma \bar{z}_a^w + \nu_i) y_{ijk}}}{\prod_{k=1}^{m_{ij}} \left[ 1 + e^{\underset{\sim}{x}'_{ijk} \underset{\sim}{\beta}_{(0)} + \sigma \bar{z}_a^w + \nu_i} \right]} \Big( \Phi(a) - \Phi(a-1) \Big) \right\}
$$

$$
\times \left( \frac{1}{\sqrt{2\pi\delta^2}} \right)^l \exp\left\{ -\sum_{i=1}^{l} \frac{(\nu_i - \beta_0)^2}{2\delta^2} \right\} \frac{1}{(1+\sigma^2)^2} \frac{1}{(1+\delta^2)^2}
$$

Similarly, let $z_i^\nu = \frac{\nu_i - \beta_0}{\delta}$ and $\bar{z}_b^\nu = \frac{p_b - p_{b-1}}{2}$, $b = 1, \ldots, 100$. We use the midpoint rule to approximate the definite integral with respect to $\underset{\sim}{\nu}$ and then get the posterior density of $\underset{\sim}{\beta}, \sigma^2, \delta^2 | \underset{\sim}{y}$

$$
\pi(\underset{\sim}{\beta}, \sigma^2, \delta^2 | \underset{\sim}{y}) \approx \prod_{i=1}^{\ell} \left\{ \sum_{b=1}^{100} \left[ \prod_{j=1}^{n_i} \left( \sum_{a=1}^{100} \frac{e^{\sum_{k=1}^{m_{ij}} (\underset{\sim}{x}'_{ijk} \underset{\sim}{\beta}_{(0)} + \beta_0 + \bar{z}_a^w \delta + \bar{z}_b^\nu \sigma) y_{ijk}}}{\prod_{k=1}^{m_{ij}} \left[ 1 + e^{\underset{\sim}{x}'_{ijk} \underset{\sim}{\beta}_{(0)} + \beta_0 + \bar{z}_a^w \delta + \bar{z}_b^\nu \sigma} \right]} \Delta(\Phi(p_a)) \right) \right] \Delta(\Phi(p_b)) \right\}
$$

$$
\times \frac{1}{(1+\sigma^2)^2} \frac{1}{(1+\delta^2)^2}.
$$

We propose to draw samples from $\underset{\sim}{\beta}, \sigma^2, \delta^2$ jointly by applying M-H sampler. Target function is $\pi(\underset{\sim}{\beta}, \sigma^2, \delta^2 | \underset{\sim}{y})$. We set the proposal function as

$$
\begin{pmatrix} \underset{\sim}{\beta} \\ \log \sigma^2 \\ \log \delta^2 \end{pmatrix} | \underset{\sim}{y} \sim \text{Normal} \left\{ \begin{pmatrix} \bar{\underset{\sim}{\beta}}_a \\ \log \bar{\sigma}_a^2 \\ \log \bar{\delta}_a^2 \end{pmatrix}, \sigma_t^2 \Sigma_a \right\},
$$

where $\frac{t}{\sigma_t^2} \sim \chi_t^2$, Chi-square on $t$ degree of freedom, i.e. $\begin{pmatrix} \beta \\ \log \sigma^2 \\ \log \delta^2 \end{pmatrix} | y \sim$ Student's t. Here $t$ is tuning constant.

We also use M-H sampler draw samples for $\nu$ and $w$ respectively. Proposal functions are $\pi_a(\nu \mid \beta_{(0)}, \beta_0, \sigma^2, y)$ and $\pi_a(w \mid \nu, \beta_{(0)}, \sigma^2, y)$ respectively from the INNA method. The target function to draw $\nu$ is

$$\pi(\nu | \beta, \sigma^2, \delta^2, y) \propto \prod_{i=1}^{\ell} \prod_{j=1}^{n_i} \left\{ \sum_{a=1}^{100} \frac{e^{\sum_{k=1}^{m_{ij}} (x'_{ijk}\beta_{(0)} + \sigma \bar{z}_a^w + \nu_i) y_{ijk}}}{\prod_{k=1}^{m_{ij}} \left[ 1 + e^{x'_{ijk}\beta_{(0)} + \sigma \bar{z}_a^w + \nu_i} \right]} \Big( \Phi(a) - \Phi(a-1) \Big) \right\}$$

$$\times \left( \frac{1}{\sqrt{2\pi\delta^2}} \right)^l \exp\left\{ -\sum_{i=1}^{l} \frac{(\nu_i - \beta_0)^2}{2\delta^2} \right\}.$$

After we get samples for $\nu$, we can use M-H sampler to draw

$$\pi(w | \nu, \beta_{(0)}, \sigma^2, y) \propto \prod_{i=1}^{\ell} \prod_{j=1}^{n_i} \prod_{k=1}^{m_{ij}} \left[ \frac{e^{(x'_{ijk}\beta_{(0)} + w_{ij}) y_{ijk}}}{1 + e^{x'_{ijk}\beta_{(0)} + w_{ij}}} \right] \times \left( \frac{1}{\sqrt{2\pi\sigma^2}} \right)^n \exp\left\{ -\sum_{i=1}^{l} \sum_{j=1}^{n_i} \frac{(w_{ij} - \nu_i)^2}{2\sigma^2} \right\}.$$

## Appendix B. Proof of Theorem 1

**Proof.** By Lemma 2, the posterior density is logconcave. Then according to Lemma 1, the posterior distribution $\tau | y$ is approximately a multivariate normal distribution.

By Lemma 1, evaluating all quantities at $\tau^*$, the mean is

$$\begin{pmatrix} \mu_w \\ \mu_\beta \end{pmatrix} = \tau^* - H^{-1} g = \begin{pmatrix} w^* \\ \beta_{(0)}^* \end{pmatrix} + \begin{pmatrix} E & F' \\ F & G \end{pmatrix} \begin{pmatrix} g_1 \\ g_2 \end{pmatrix} = \begin{pmatrix} w^* + E g_1 + F' g_2 \\ \beta_{(0)}^* + F g_1 + G g_2 \end{pmatrix}.$$

Also, the covariance matrix is

$$-H^{-1} = \begin{pmatrix} D & C' \\ C & B \end{pmatrix}^{-1} = \begin{pmatrix} E & F' \\ F & G \end{pmatrix}.$$

Therefore, by Lemma 1, the approximate joint posterior density of $w, \beta_{(0)} | y$ is

$$\begin{pmatrix} w \\ \beta_{(0)} \end{pmatrix} | y \sim \text{Normal}\left\{ \begin{pmatrix} \mu_w \\ \mu_\beta \end{pmatrix}, \begin{pmatrix} E & F' \\ F & G \end{pmatrix} \right\}.$$

Finally, using the property of the multivariate normal density, the conditional posterior density of $w | \beta_{(0)}, y$ and $\beta_{(0)} | y$ can also be approximated by multivariate normal distributions,

$$w | \beta_{(0)}, y \sim \text{Normal}\{ \mu_w - D^{-1} C'(\beta_{(0)} - \mu_\beta), D^{-1} \} \text{ and } \beta_{(0)} | y \sim \text{Normal}\{ \mu_\beta, G \},$$

where

$$\mu_w = w^* + E g_1 + F' g_2 \text{ and } \mu_\beta = \beta_{(0)}^* + F g_1 + G g_2.$$

$\square$

## Appendix C. Proof of Theorem 2

**Proof.** First, look at the exponent terms containing $w$ in the above approximate posterior density function

$$\left[\underset{\sim}{w} - \left(\underset{\sim}{\mu}_w - D^{-1}C'(\underset{\sim}{\beta}_{(0)} - \underset{\sim}{\mu}_\beta)\right)\right]'D\left[\underset{\sim}{w} - \left(\underset{\sim}{\mu}_w - D^{-1}C'(\underset{\sim}{\beta}_{(0)} - \underset{\sim}{\mu}_\beta)\right)\right] + \left[\underset{\sim}{w} - \underset{\sim}{\mu}_v\right]'(\sigma^2 I)^{-1}\left[\underset{\sim}{w} - \underset{\sim}{\mu}_v\right]$$

$$= \Sigma'_{\underset{\sim}{w}}(D + \frac{1}{\sigma^2}I)\Sigma_{\underset{\sim}{w}} + \left[\underset{\sim}{\mu}_w - D^{-1}C'\left(\underset{\sim}{\beta}_{(0)} - \underset{\sim}{\mu}_\beta\right) - \underset{\sim}{\mu}_v\right]'(D^{-1} + \sigma^2 I)^{-1}\left[\underset{\sim}{\mu}_w - D^{-1}C'\left(\underset{\sim}{\beta}_{(0)} - \underset{\sim}{\mu}_\beta\right) - \underset{\sim}{\mu}_v\right],$$

where $\Sigma_{\underset{\sim}{w}} = \left[\underset{\sim}{w} - (D + \frac{1}{\sigma^2}I)^{-1}\left(D\underset{\sim}{\mu}_w - C'(\underset{\sim}{\beta}_{(0)} - \underset{\sim}{\mu}_\beta) + \frac{1}{\sigma^2}\underset{\sim}{\mu}_v\right)\right]$.

Then it can show that the $\pi_a(\underset{\sim}{w} \mid \underset{\sim}{v}, \underset{\sim}{\beta}_{(0)}, \sigma^2, \underset{\sim}{y})$ is

$$\underset{\sim}{w}|\underset{\sim}{v}, \underset{\sim}{\beta}_{(0)}, \sigma^2, \underset{\sim}{y} \stackrel{\text{app}}{\sim} \text{Normal}\left\{(D + \frac{1}{\sigma^2}I)^{-1}\left(D\underset{\sim}{\mu}_w - C'(\underset{\sim}{\beta}_{(0)} - \underset{\sim}{\mu}_\beta) + \frac{1}{\sigma^2}\underset{\sim}{\mu}_v\right), (D + \frac{1}{\sigma^2}I)^{-1}\right\}.$$

Notice that $(D + \frac{1}{\sigma^2}I)$ is diagonal matrix. Then given $\underset{\sim}{v}, \underset{\sim}{\beta}_{(0)}, \sigma^2, \underset{\sim}{y}$, all $w_{ij}$s are independent. This is an important result because parallel computation can be done for $w_{ij}$, which accommodates time-consuming and massive storage challenges in big data analysis. This result holds for the exact conditional posterior density of the $\mu_{ij}$. Since $\underset{\sim}{w}$ has a multivariate normal distribution, we can integrate out $\underset{\sim}{w}$ from the joint approximate posterior density $\pi_a(\underset{\sim}{w}, \underset{\sim}{v}, \underset{\sim}{\beta}, \sigma^2, \delta^2 \mid \underset{\sim}{y})$, and obtain the joint posterior density of $\underset{\sim}{v}, \underset{\sim}{\beta}, \sigma^2$ and $\delta^2$

$$\pi_a(\underset{\sim}{v}, \underset{\sim}{\beta}, \sigma^2, \delta^2 \mid \underset{\sim}{y}) \propto e^{-\frac{1}{2}\left\{\left[\underset{\sim}{\mu}_v - D^{-1}C'\left(\underset{\sim}{\beta}_{(0)} - \underset{\sim}{\mu}_\beta\right) - \underset{\sim}{\mu}_w\right]'(D^{-1} + \sigma^2 I)^{-1}\left[\underset{\sim}{\mu}_v - D^{-1}C'\left(\underset{\sim}{\beta}_{(0)} - \underset{\sim}{\mu}_\beta\right) - \underset{\sim}{\mu}_w\right]\right\}}$$

$$\times e^{-\frac{1}{2}\left\{\left[\underset{\sim}{v} - \beta_0\underset{\sim}{j}\right]'(\delta^2 I)^{-1}\left[\underset{\sim}{v} - \beta_0\underset{\sim}{j}\right] + \left[\underset{\sim}{\beta}_{(0)} - \underset{\sim}{\mu}_\beta\right]'G^{-1}\left[\underset{\sim}{\beta}_{(0)} - \underset{\sim}{\mu}_\beta\right]\right\}}$$

$$\times \frac{|D|^{1/2}}{|\delta^2 I|^{1/2}|D + \frac{1}{\sigma^2}I|^{1/2}|G|^{1/2}} \frac{1}{(1 + \sigma^2)^2} \frac{1}{(1 + \delta^2)^2}.$$

Next, we will show that the approximate conditional posterior density of $v_i$ is also normal distribution and all $v_i$s are independent as well. Here we consider each $v_i$. Let $\sum_{i=1}^{\ell}\sum_{j=1}^{n_i} = n$, $(\sigma_{ij}^2)_{n \times n} = D^{-1} + \sigma^2 I$, $(\underset{\sim}{t}_{ij})_{n \times 1} = D^{-1}C$ and $(\mu_{w_{ij}})_{n \times 1} = \underset{\sim}{\mu}_w$.

Look at the exponent only containing $v_i, i = 1, \ldots, \ell$. in the $\pi_a(\underset{\sim}{v}, \underset{\sim}{\beta}, \sigma^2, \delta^2 \mid \underset{\sim}{y})$

$$\sum_{i=1}^{\ell}\sum_{j=1}^{n_i}\frac{1}{\sigma_{ij}^2}\left[v_i - \mu_{w_{ij}} + \underset{\sim}{t}'_{ij}(\underset{\sim}{\beta}_{(0)} - \underset{\sim}{\mu}_\beta)\right]^2 + \frac{1}{\delta^2}\sum_{i=1}^{\ell}(v_i - \beta_0)^2$$

$$= \sum_{i=1}^{\ell}(\frac{1}{\sum_{j=1}^{n_i}\sigma_{ij}^2} + \frac{1}{\delta^2})^{-1}\left\{v_i - \frac{\left(\frac{1}{\sum_{j=1}^{n_i}\sigma_{ij}^2}\right)\left[\bar{\mu}_{w_i} - \underset{\sim}{\bar{t}}'_i(\underset{\sim}{\beta}_{(0)} - \underset{\sim}{\mu}_\beta)\right] + \frac{1}{\delta^2}\beta_0}{\frac{1}{\sum_{j=1}^{n_i}\sigma_{ij}^2} + \frac{1}{\delta^2}}\right\}^2$$

$$+ \sum_{i=1}^{\ell}\left(\frac{1}{1/\sum_{j=1}^{n_i}\sigma_{ij}^2} + \delta^2\right)^{-1}\left\{\bar{\mu}_{w_i} - \underset{\sim}{\bar{t}}'_i(\underset{\sim}{\beta}_{(0)} - \underset{\sim}{\mu}_\beta) - \beta_0\right\}^2$$

$$+ \sum_{i=1}^{\ell}\sum_{j=1}^{n_i}\frac{1}{\sigma_{ij}^2}\left\{(\underset{\sim}{\bar{t}}_i - \underset{\sim}{t}_{ij})'(\underset{\sim}{\beta}_{(0)} - \underset{\sim}{\mu}_\beta) - (\bar{\mu}_{w_i} - \mu_{w_{ij}})\right\}^2,$$

where $\bar{\mu}_{w_i} = \frac{1}{n_i}\sum_{j=1}^{n_i}\mu_{w_{ij}}$ and $\underset{\sim}{\bar{t}}_i = \frac{1}{n_i}\sum_{j=1}^{n_i}\underset{\sim}{t}_{ij}$.

Then it is easy to see that

$$v_i|\beta_0, \sigma^2, \delta^2, \underset{\sim}{y} \stackrel{\text{app}}{\sim} \text{Normal}\left\{\frac{\left(\frac{1}{\sum_{j=1}^{n_i}\sigma_{ij}^2}\right)\left[\bar{\mu}_{w_i} - \underset{\sim}{\bar{t}}'_i(\underset{\sim}{\beta}_{(0)} - \underset{\sim}{\mu}_\beta)\right] + \frac{1}{\delta^2}\beta_0}{\frac{1}{\sum_{j=1}^{n_i}\sigma_{ij}^2} + \frac{1}{\delta^2}}, \frac{1}{\sum_{j=1}^{n_i}\sigma_{ij}^2} + \frac{1}{\delta^2}\right\}.$$

Similarly, we can use parallel computing to draw $\nu_i$, $i = 1, \ldots, \ell$ as well since all of them are independent given $\underset{\sim}{\beta}_{(0)}, \beta_0, \sigma^2, \delta^2$. Then we can integrate out $\underset{\sim}{\nu}$ from the joint approximate posterior density $\pi_a(\underset{\sim}{\nu}, \underset{\sim}{\beta}, \sigma^2, \delta^2 \mid \underset{\sim}{y})$ and obtain the joint posterior density of $\underset{\sim}{\beta}, \sigma^2$ and $\delta^2$

$$
\begin{aligned}
\pi_a(\underset{\sim}{\beta}, \sigma^2, \delta^2 | \underset{\sim}{y}) \propto \exp &\left\{ -\frac{1}{2} \sum_{i=1}^{\ell} \left( \frac{1}{1/\sum_{j=1}^{n_i} \sigma_{ij}^2} + \delta^2 \right)^{-1} \left[ \bar{\mu}_{w_i} - \underset{\sim}{\bar{t}}_i'(\underset{\sim}{\beta}_{(0)} - \underset{\sim}{\mu}_\beta) - \beta_0 \right]^2 \right\} \\
&\times \exp \left\{ -\frac{1}{2} \sum_{i=1}^{\ell} \sum_{j=1}^{n_i} \frac{1}{\sigma_{ij}^2} \left[ \{ (\underset{\sim}{\bar{t}}_i - \underset{\sim}{t}_{ij})'(\underset{\sim}{\beta}_{(0)} - \underset{\sim}{\mu}_\beta) - (\bar{\mu}_{w_i} - \mu_{w_{ij}}) \right]^2 \right\} \\
&\times \prod_{i=1}^{l} \left( \frac{1}{\sum_{j=1}^{n_i} \sigma_{ij}^2} + \frac{1}{\delta^2} \right)^{\frac{1}{2}} \frac{1}{|\delta^2 I|^{1/2}|D + \frac{1}{\sigma^2} I|^{1/2}} \frac{1}{(1+\sigma^2)^2} \frac{1}{(1+\delta^2)^2} \\
&= e^{-\frac{1}{2}\left( \underset{\sim}{\mu}_w - D^{-1}C'(\underset{\sim}{\beta}_{(0)} - \underset{\sim}{\mu}_\beta) - \beta_0 \underset{\sim}{j} \right)'\left( D^{-1} + \sigma^2 I + \delta^2 I \right)^{-1}\left( \underset{\sim}{\mu}_w - D^{-1}C'(\underset{\sim}{\beta}_{(0)} - \underset{\sim}{\mu}_\beta) - \beta_0 \underset{\sim}{j} \right)} \\
&\times e^{-\frac{1}{2}\left( \underset{\sim}{\beta}_{(0)} - \underset{\sim}{\mu}_\beta \right)' G^{-1}\left( \underset{\sim}{\beta}_{(0)} - \underset{\sim}{\mu}_\beta \right)} \\
&\times \prod_{i=1}^{l} \left( \frac{1}{\sum_{j=1}^{n_i} \sigma_{ij}^2} + \frac{1}{\delta^2} \right)^{\frac{1}{2}} \frac{1}{|\delta^2 D + \frac{\delta^2}{\sigma^2} I|^{1/2}} \frac{1}{(1+\sigma^2)^2} \frac{1}{(1+\delta^2)^2}.
\end{aligned}
$$

Next we assume that the conditional posterior density of $\underset{\sim}{\beta}|\sigma^2, \delta^2, \underset{\sim}{y}$ has an approximate multivariate normal density,

$$
\left( \begin{array}{c} \beta_0 \\ \underset{\sim}{\beta}_{(0)} \end{array} \right) |\sigma^2, \delta^2, \underset{\sim}{y} \sim \text{Normal}\left\{ \left( \begin{array}{c} \omega_0 \\ \underset{\sim}{\omega}_{(0)} \end{array} \right), \left( \begin{array}{cc} \delta_0^2 & \gamma' \\ \underset{\sim}{\gamma} & \Delta_{(0)} \end{array} \right)^{-1} \right\},
$$

which is denoted by $\pi_a(\underset{\sim}{\beta} \mid \sigma^2, \delta^2, \underset{\sim}{y})$. The density function is

$$
\pi_a(\underset{\sim}{\beta} \mid \sigma^2, \delta^2, \underset{\sim}{y}) \propto \left| \left( \begin{array}{cc} \delta_0^2 & \gamma' \\ \underset{\sim}{\gamma} & \Delta_{(0)} \end{array} \right) \right|^{\frac{1}{2}} \times e^{-\frac{1}{2}\left( \begin{array}{c} \beta_0 - \omega_0 \\ \underset{\sim}{\beta}_{(0)} - \underset{\sim}{\omega}_{(0)} \end{array} \right)'\left( \begin{array}{cc} \delta_0^2 & \gamma' \\ \underset{\sim}{\gamma} & \Delta_{(0)} \end{array} \right)\left( \begin{array}{c} \beta_0 - \omega_0 \\ \underset{\sim}{\beta}_{(0)} - \underset{\sim}{\omega}_{(0)} \end{array} \right)}
$$

So the exponent terms are

$$
\left( \begin{array}{c} \beta_0 - \omega_0 \\ \underset{\sim}{\beta}_{(0)} - \underset{\sim}{\omega}_{(0)} \end{array} \right)'\left( \begin{array}{cc} \delta_0^2 & \gamma' \\ \underset{\sim}{\gamma} & \Delta_{(0)} \end{array} \right)\left( \begin{array}{c} \beta_0 - \omega_0 \\ \underset{\sim}{\beta}_{(0)} - \underset{\sim}{\omega}_{(0)} \end{array} \right).
$$

Consider the exponent terms containing $\underset{\sim}{\beta}_{(0)}$ and $\beta_0$

$$
\begin{aligned}
&\left( \underset{\sim}{\mu}_w - D^{-1}C'(\underset{\sim}{\beta}_{(0)} - \underset{\sim}{\mu}_\beta) - \beta_0 \underset{\sim}{j} \right)'\left( D^{-1} + \sigma^2 I + \delta^2 I \right)^{-1}\left( \underset{\sim}{\mu}_w - D^{-1}C'(\underset{\sim}{\beta}_{(0)} - \underset{\sim}{\mu}_\beta) - \beta_0 \underset{\sim}{j} \right) \\
&\quad + \left( \underset{\sim}{\beta}_{(0)} - \underset{\sim}{\mu}_\beta \right)' G^{-1}\left( \underset{\sim}{\beta}_{(0)} - \underset{\sim}{\mu}_\beta \right) \\
&= \underset{\sim}{\beta}_{(0)}'\left[ CD^{-1}(D^{-1} + \sigma^2 I + \delta^2 I)^{-1}D^{-1}C' + G^{-1} \right]\underset{\sim}{\beta}_{(0)} + \underset{\sim}{j}'(D^{-1} + \sigma^2 I + \delta^2 I)^{-1}\underset{\sim}{j}\beta_0^2 \\
&\quad - 2\left[ (\underset{\sim}{\mu}_w + D^{-1}C'\underset{\sim}{\mu}_\beta)'(D^{-1} + \sigma^2 I + \delta^2 I)D^{-1}C' + \underset{\sim}{\mu}_\beta' G^{-1} \right]\underset{\sim}{\beta}_{(0)} \\
&\quad - 2\left[ (\underset{\sim}{\mu}_w + D^{-1}C'\underset{\sim}{\mu}_\beta)'(D^{-1} + \sigma^2 I + \delta^2 I)^{-1}\underset{\sim}{j} \right]\beta_0 + 2CD^{-1}(D^{-1} + \sigma^2 I + \delta^2 I)^{-1}\underset{\sim}{j}\beta_0\underset{\sim}{\beta}_{(0)} \\
&\quad + (D^{-1}C'\underset{\sim}{\mu}_\beta + \underset{\sim}{\mu}_w)'(D^{-1} + \sigma^2 I + \delta^2 I)^{-1}\underset{\sim}{j}\beta_0\underset{\sim}{\beta}_{(0)} \\
&\quad (\underset{\sim}{\mu}_w + D^{-1}C'\underset{\sim}{\mu}_\beta)'(D^{-1} + \sigma^2 I + \delta^2 I)^{-1}(\underset{\sim}{\mu}_w + D^{-1}C'\underset{\sim}{\mu}_\beta) + \underset{\sim}{\mu}_\beta' G^{-1}\underset{\sim}{\mu}_\beta.
\end{aligned}
$$

We know those two exponent parts are equal, so we have

$$\Delta_{(0)} = CD^{-1}(D^{-1} + \sigma^2 I + \delta^2 I)^{-1} D^{-1} C' + G^{-1},$$

$$\delta_0^2 = \underset{\sim}{j}'(D^{-1} + \sigma^2 I + \delta^2 I)^{-1}\underset{\sim}{j},$$

$$\underset{\sim}{\gamma} = CD^{-1}(D^{-1} + \sigma^2 I + \delta^2 I)^{-1}\underset{\sim}{j},$$

$$\begin{pmatrix} \omega_0 \\ \underset{\sim}{\omega}_{(0)} \end{pmatrix} = \begin{pmatrix} \delta_0^2 & \underset{\sim}{\gamma}' \\ \underset{\sim}{\gamma} & \Delta_{(0)} \end{pmatrix}^{-1} \begin{pmatrix} (\underset{\sim}{\mu}_w + D^{-1}C'\underset{\sim}{\mu}_\beta)'(D^{-1} + \sigma^2 I + \delta^2 I)^{-1}\underset{\sim}{j} \\ (\underset{\sim}{\mu}_w + D^{-1}C'\underset{\sim}{\mu}_\beta)'(D^{-1} + \sigma^2 I + \delta^2 I)D^{-1}C' + \underset{\sim}{\mu}_\beta' G^{-1} \end{pmatrix}.$$

That is, $\underset{\sim}{\beta}|\sigma^2, \delta^2, \underset{\sim}{y}$ approximately follows multivariate normal distribution,

$$\begin{pmatrix} \beta_0 \\ \underset{\sim}{\beta}_{(0)} \end{pmatrix} | \sigma^2, \delta^2, \underset{\sim}{y} \sim \text{Normal}\left\{ \begin{pmatrix} \omega_0 \\ \underset{\sim}{\omega}_{(0)} \end{pmatrix}, \begin{pmatrix} \delta_0^2 & \underset{\sim}{\gamma}' \\ \underset{\sim}{\gamma} & \Delta_{(0)} \end{pmatrix}^{-1} \right\},$$

Then we can easily integrate out $\underset{\sim}{\beta}$ from the joint density of $\underset{\sim}{\beta}, \sigma^2, \delta^2 | \underset{\sim}{y}$, and get the posterior density of $\sigma^2, \delta^2 | \underset{\sim}{y}$

$$\pi_a(\sigma^2, \delta^2 | \underset{\sim}{y}) \propto \left| \begin{matrix} \delta_0^2 & \underset{\sim}{\gamma}' \\ \underset{\sim}{\gamma} & \Delta_{(0)} \end{matrix} \right|^{-\frac{1}{2}} \times \prod_{i=1}^{l} \left( \frac{1}{\sum_{j=1}^{n_i} \sigma_{ij}^2} + \frac{1}{\delta^2} \right)^{\frac{1}{2}} \frac{1}{|\delta^2 D + \frac{\delta^2}{\sigma^2} I|^{1/2}} \frac{1}{(1+\sigma^2)^2} \frac{1}{(1+\delta^2)^2}$$

$$\times \exp\left\{ -\frac{1}{2}(\underset{\sim}{\mu}_w + D^{-1}C'\underset{\sim}{\mu}_\beta)'(D^{-1} + \sigma^2 I + \delta^2 I)^{-1}(\underset{\sim}{\mu}_w + D^{-1}C'\underset{\sim}{\mu}_\beta) + \underset{\sim}{\mu}_\beta' G^{-1}\underset{\sim}{\mu}_\beta \right\}$$

$$\times \exp\left\{ -\frac{1}{2} \begin{pmatrix} \beta_0 - \omega_0 \\ \underset{\sim}{\beta}_{(0)} - \underset{\sim}{\omega}_{(0)} \end{pmatrix}' \begin{pmatrix} \delta_0^2 & \underset{\sim}{\gamma}' \\ \underset{\sim}{\gamma} & \Delta_{(0)} \end{pmatrix} \begin{pmatrix} \beta_0 - \omega_0 \\ \underset{\sim}{\beta}_{(0)} - \underset{\sim}{\omega}_{(0)} \end{pmatrix} \right\}.$$

$\square$

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
