# Peer review of "Bayesian Logistic Regression Model for Sub-Areas"

_stats, doi:10.3390/stats6010013_

Round 1

Reviewer 1 Report

Thanks for the review opportunity. Overall, this is a very well written manuscript with significant novel contribution to the field. I have some minor comments in the following. 

1. Line 225: please define abbreviation PSU. 

2. Line 307: "... Based the updated likelihood.." missing a on between "Based" and "the"

3. Can this approximation method "INNA" be easily extended to a more general situation beyond the two-fold hierarchical Bayesian logistic regression model discussed in the manuscript? If so, please provide a short paragraph to explain. 

4.It seems that INNA tends to have smaller PSD than exact method in Figure 5. Can the author please comment on this? Is it because the same function in INNA was used as proposal function in the M-H sampler? 

Author Response

Response to Reviewer 1 Comments

Thanks for the review opportunity. Overall, this is a very well written manuscript with significant novel contribution to the field. I have some minor comments in the following. 

  1. Line 225: please define abbreviation PSU. 

Response: Corrected in the manuscript. PSU refer to “Primary Sampling Units”

  1. Line 307: "... Based the updated likelihood.." missing a on between "Based" and "the"

Response: Corrected in the manuscript.

  1. Can this approximation method "INNA" be easily extended to a more general situation beyond the two-fold hierarchical Bayesian logistic regression model discussed in the manuscript? If so, please provide a short paragraph to explain. 

Response: Agreed. We added a paragraph in the conclusion section to discuss the application of the INNA method. It is not only for two-fold HB logistic regression model but all the HB logistic regression model with latent Gaussian field.

  1. It seems that INNA tends to have smaller PSD than exact method in Figure 5. Can the author please comment on this? Is it because the same function in INNA was used as proposal function in the M-H sampler? 

       Response: The difference between exact method and INNA method are not statistically significant. In the INNA method, we find a convenient point to expand the log-likelihood in a second-order multivariate Taylor’s series expansion and made the approximation rather than fining mode for each point. Such approximation lead the slightly smaller PSD when comparing with the exact method.

Reviewer 2 Report

The paper is interesting and its content deserves publication. However, I am afraid that the manuscript, in its present form, does not help the reader to understand its main contribution, and therefore I suggest that the Author works for a clearer presentation.

From one side, the manuscript refers to the applied case study of the Nepal Living Standard Survey, from the other side, in its Section 3, it appears very theoretical. The two features are very difficult to maintain.

The fundamental Section 3, in order to be appreciated by a non-specialist reader, ought to be converted into a more colloquial text if the Author likes to propose a work that can be read with more ease.

Perhaps the Author might start from the INNA model of pages 8 and 9, i.e. what now is Theorem 2,  and illustrate how it is achieved…

There are some other difficulties.

As an example, the difference between the first formula after line 144 and the first one after line 163 is in the suffixes, but is difficult to see this in the formulae, without further explanation.

Also the statements of the Subarea Hierarchical Logistic Regression Model, contained in the shorter Section 2, may be presented in a more colloquial way. (Note for instance that the joint  prior π(˜β, δ2, σ2) seems to appear twice in the section, in very near points).

More important: why Section 2 does contain the sentences:

“The model and method we proposed for many small areas and sub-areas is not only for our application on NLSS II. It can be applied to other population-based surveys with binary responses. In our application, we have binary data (good health versus poor health) for each individual within a household, and these households are within wards.” ?

They refer to the application, while Section 2 (and also Section 3) are presented as theoretical and very general.

The English ought to be carefully checked throughout the whole paper, with special reference to

a) the use of the article “the”,

b) the singular/plural (at this regard please look, as examples, at line 141, also at lines 45-46, and line 47)

But there are also further minor points, as:

is the Author confident that the expressions “one-fold”, “two-fold”, “expit” are familiar to any reader?

Regarding only lines 1-188, some of such minor points are listed

Line 29: “we are particular interested”?

Line 32: the word “consistency” may be dangerous if the statistical jargon is used…

Line 72: is “this logistic regression model” the one proposed in Nandram, Chen, Shu and Binod (2018)?

Line 79: the definition of Scott et al. (2013) is ONE among several definition of “big data”…

Lines 84-85: perhaps “uses A stratification …”

Line 112: computation

Line 118: “concluding REmarks”?

Line 140: definite and not defenite

Line 141: is the expression “we also use certain point …” Correct?

End of page 5: first and not frist

5-th line of page 6: Taylor and not Talor

10-th line of page 6: Is “Then solve for…, we can easily get….” Correct?

12-th line of page 6: It begins with “Second”, but where is “First”?

18-th line of page 6: “Solve for…” or “Solving for…”?

Line 166: The text mentions the identity matrix I, but is the Author sure that such matrix appears in the formulae nearby?

Similar (but several) minor points can be raised in the other parts of the work.

In the heading of Tables 4 and 5 the expression “five number summaries” appears. Is this conveniently mentioned in the text? Are the 5 numbers “only” Min, Q1, Mean, Q3 and Max?

Reviewer 3 Report

The manuscript presents an application of the hierarchical bayesian logistic regression model and uses an Integrated nested normal approximation method for posterior approximation. The manuscript is generally well-written and contains detailed explanations of the methodologies,  and well-documented numerical study processes and results. 

However, upon checking some of the manuscript’s citations, I found one of the authors’ previous papers (Nandram, Chen, Shu and Binod (2018) Bayesian Logistic Regression for Small Areas with Numerous Households. In Statistics and Application 2018) very similar to the current work. Comparing the two papers shows many signs of paraphrasing without proper citation, and the incremental contribution from the previous paper is not significant.

I suggest the authors focus on incremental contributions when writing, and cite the previously published paper in all the locations where previous contents are used/paraphrased.

Author Response

Response to Reviewer 3 Comments

The manuscript presents an application of the hierarchical bayesian logistic regression model and uses an Integrated nested normal approximation method for posterior approximation. The manuscript is generally well-written and contains detailed explanations of the methodologies, and well-documented numerical study processes and results. 

 However, upon checking some of the manuscript’s citations, I found one of the authors’ previous papers (Nandram, Chen, Shu and Binod (2018) Bayesian Logistic Regression for Small Areas with Numerous Households. In Statistics and Application 2018) very similar to the current work. Comparing the two papers shows many signs of paraphrasing without proper citation, and the incremental contribution from the previous paper is not significant.

I suggest the authors focus on incremental contributions when writing, and cite the previously published paper in all the locations where previous contents are used/paraphrased.

Response: We agreed that we did not mention clearly about Nandram et al. (2018) paper in this manuscript. In order to make things clear, we added one paragraph in the introduction section. This manucript is an extension to the one-fold area level logistric regression model proposed in Nandram et al. (2018). Nandram et al. (2018) proposed a one-fold hierarchical Bayesian logistic regression model and apply the model to NLSS II data. The main object is to make inference for the finite population proportion of individuals with a specific character for each area. However, the one-fold model ignores the sub-area level structure in the data. As an extension of Nandram et al. (2018), we are particularly interested in small area models that can capture the hierarchical structure of the NLSS II data in this paper. Although the one-fold basic models are very popular and in common use in producing reliable estimates, the hierarchical structure of the data and the consistency between the estimates for different levels may not hold. In particular, the sampling designs of many population-based survey were two-stage stratified sampling as NLSS II. But if we use one-fold unit level model to fit the data, sub-area level effects would be ignored. On the other hand, the INNA computation method is also an extension to the INNA method proposed in Nandram et al. (2018). In this manuscript, we discuss the method for the sub-area level model which is more complicated. The general ideas are similar but the equations and details are different. The subarea level model is better for the data containing the subarea structure such as the NLSS II data.

Round 2

Reviewer 2 Report

The Authors made a very careful revision of the first version of the manuscript. I suggest to publish this new version.